# Parvovirus nonstructural protein 2 interacts with chromatin-regulating cellular proteins

**Salla Mattola**[1], **Kari Salokas**[2], **Vesa Aho**[1], **Elina Mäntylä**[3], **Sami Salminen**[1], **Satu Hakanen**[1], **Einari A. Niskanen**[4], **Julija Svirskaite**[2], **Teemu O. Ihalainen**[3], **Kari J. Airenne**[5], **Minna Kaikkonen-Määttä**[4], **Colin R. Parrish**[6], **Markku Varjosalo**[2], **Maija Vihinen-Ranta**[1] *

**1** Department of Biological and Environmental Science and Nanoscience Center, University of Jyvaskyla, Jyvaskyla, Finland, **2** Institute of Biotechnology and Helsinki Institute of Life Science (HiLIFE), University of Helsinki, Helsinki, Finland, **3** BioMediTech, Faculty of Medicine and Health Technology, Tampere University, Tampere, Finland, **4** Institute of Biomedicine, University of Eastern Finland, Kuopio, Finland, **5** Kuopio Center for Gene and Cell Therapy (KCT), Kuopio, Finland, **6** Baker Institute for Animal Health, Department of Microbiology and Immunology, College of Veterinary Medicine, University of Cornell, Ithaca, New York, United States of America

explanation These authors contributed equally to this work.
* maija.vihinen-ranta@jyu.fi

**Data Availability Statement:** All relevant data are included in the manuscript and its Supporting information files.

## Abstract

Autonomous parvoviruses encode at least two nonstructural proteins, NS1 and NS2. While NS1 is linked to important nuclear processes required for viral replication, much less is known about the role of NS2. Specifically, the function of canine parvovirus (CPV) NS2 has remained undefined. Here we have used proximity-dependent biotin identification (BioID) to screen for nuclear proteins that associate with CPV NS2. Many of these associations were seen both in noninfected and infected cells, however, the major type of interacting proteins shifted from nuclear envelope proteins to chromatin-associated proteins in infected cells. BioID interactions revealed a potential role for NS2 in DNA remodeling and damage response. Studies of mutant viral genomes with truncated forms of the NS2 protein suggested a change in host chromatin accessibility. Moreover, further studies with NS2 mutants indicated that NS2 performs functions that affect the quantity and distribution of proteins linked to DNA damage response. Notably, mutation in the splice donor site of the NS2 led to a preferred formation of small viral replication center foci instead of the large coalescent centers seen in wild-type infection. Collectively, our results provide insights into potential roles of CPV NS2 in controlling chromatin remodeling and DNA damage response during parvoviral replication.

## Author summary

Parvoviruses are small, nonenveloped DNA viruses, that besides being noteworthy pathogens in many animal species, including humans, are also being developed as vectors for gene and cancer therapy. Canine parvovirus is an autonomously replicating parvovirus that encodes two nonstructural proteins, NS1 and NS2. NS1 is required for viral DNA

**Funding:** This work was financed by the Jane and Aatos Erkko Foundation (MVR); Academy of Finland under the award number 330896 (MVR); Biocenter Finland, viral gene transfer (MVR), and the Graduate School of the University of Jyvaskyla (SM). The funders had no role in study design, data collection and analysis, decision to publish, or preparation of the manuscript.

**Competing interests:** The authors have declared that no competing interests exist.

replication and packaging, as well as gene expression. However, very little is known about the function of NS2. Our studies indicate that NS2 serves a previously undefined important function in chromatin modification and DNA damage responses. Therefore, it appears that although both NS1 and NS2 are needed for a productive infection they play very different roles in the process.

## Introduction

Autonomous parvoviruses are small single-stranded DNA viruses that depend on host cell nuclear machinery for their replication. The nuclear events of parvovirus infection include genome replication, viral assembly, and genome packaging, which require the nuclear import of structural and nonstructural viral proteins. The nonstructural protein 1 (NS1) of canine parvovirus (CPV) is a multifunctional protein with site-specific DNA binding, ATPase, nickase, and helicase activities [1,2], and its expression induces apoptosis in host cells [3,4]. NS1 is essential for initiation and direction of viral DNA replication. However, the role of the nonstructural protein 2 (NS2) in viral replication has so far remained undefined.

Autonomous parvoviruses have an ~5-kb single-stranded DNA genome encoding two viral structural proteins, VP1 and VP2, as well as two nonstructural proteins NS1 and NS2 [5]. The CPV NS2 protein is produced from the left-hand open reading frame of the viral genome and contains 87 amino-terminal amino acids that are in common with NS1 joined by mRNA splicing to 78 amino acids from an alternative open reading frame [6,7]. Previous knowledge on NS2 protein function in parvovirus replication is mainly derived from studies of minute virus of mice (MVM). MVM NS2 is required for efficient viral replication and capsid assembly [8,9]. In infected cells MVM NS2 is known to interact with two members of the 14-3-3 family of signaling proteins [10] and the survival motor neuron protein (Smn) [11]. Mutations in MVM NS2 splice acceptor or termination sites lead to severe replication defects in murine cells, whereas in other cells lines mutant viruses replicate more efficiently [12,13], suggesting that the requirement for MVM NS2 is cell-type specific. Moreover, MVM NS2 is required for the growth and development of viral replication centers [14]. Notably, MVM NS2 also interacts with the nuclear export factor CRM1 (also known as exportin1) [15,16]. CRM1 mediates the nuclear export of nuclear export signal containing proteins [17–19]. Although the detailed mechanisms of MVM capsid nuclear egress are still not well understood, the interaction between NS2 and CRM1 seems to be essential for the progeny virus capsid export [20–23]. Studies of another parvovirus closely related to CPV, feline panleucopenia virus (FPV), have shown that FPV NS2 plays a significant role in blocking pathways that promote IFN-β production, allowing the virus to evade the host antiviral innate immune response [24]. Much less is known about the role of CPV NS2 in infection. To our knowledge, the only study assessing CPV NS2 function used various NS2 mutants containing mutations and deletions that affect NS2 mRNA splicing and protein expression, or that terminate the NS2 open reading frame without altering NS1. The impact of the mutants on viral replication depended on the site of the mutation, and infection efficiency was found to be decreased with the NS2 donor mutant [6].

During the nuclear replication parvoviruses must either confront or embrace the chromatin remodeling machinery of the host cell. To ensure a productive infection, viruses have to recruit the cellular histone modifying and nucleosome remodeling machinery for the activation of the viral genome. Many viruses are able to counteract host-mediated silencing by recruiting and redirecting cellular histone remodeling proteins to enhance viral gene expression and

replication. In herpes simplex virus 1 (HSV-1) infection the viral genome is chromatinized after its nuclear entry [25]. In a productive HSV-1 infection the modification of bound histones to an active euchromatic state is promoted by viral proteins such as ICP0 and by recruiting cellular proteins [26,27]. For example, the chromatin remodeling factor SNF2H (SMARCA5) protein from ISWI family complexes facilitates the transcription of viral immediately-early genes from the HSV-1 genome by removing or remodeling histones associated with viral promoters [28]. Viruses have also developed strategies for regulating host transcription by inactivating certain aspects of chromatin modeling while exploiting others to advance the viral life cycle. Most likely, manipulation of host cellular functions can be orchestrated and tuned by viral proteins. Consistent with this model, the HSV-1 single-stranded DNA-binding protein ICP8 has been found associated with cellular proteins involved in DNA replication, DNA repair, chromatin remodeling and RNA processing [29,30]. Similar to other DNA viruses, parvovirus replication is potentially dependent on the activation state of nucleosomes present on the nuclear viral genome [31–34]. The molecular mechanisms by which parvoviral proteins are involved in the interaction and epigenetic modification of nucleosomes associated with viral genomes remain poorly understood.

The DNA damage response (DDR) machinery plays a significant role in cells by maintaining normal chromatin functions within regions of damage [35]. DDR is initiated by sensor protein-mediated detection of DNA lesions, which is followed by the activation of major signaling kinases, ataxia telangiectasia mutated (ATM), and ATM Rad3-related (ATR) and DNA-dependent protein kinase (DNA-PK). This promotes signal-transduction through a series of downstream effector molecules from phosphoinositide 3 kinase-like kinases and lead to the phosphorylation of the histone H2A variant, H2AX. DDR plays a dual role in the regulation of viral replication. DDR is also involved in the intrinsic antiviral mechanisms that counter the nuclear replication of DNA viruses [36–39]. Conversely, DDR is also activated by many DNA viruses, and DDR factors are recruited by viruses to promote viral replication [39–42]. Autonomous parvovirus infection results in the induction of cellular DNA breaks and DDR activation by ATR and ATM signaling pathways [43–46], and it is accompanied by pre-mitotic cell cycle arrest [14,43,47,48]. In MVM infection viral replication is located at cellular DNA damage sites [34,49].

In recent years proximity-dependent biotin identification (BioID) method has been increasingly used to provide fundamental insight into the protein-protein interactions of mammalian cells. These approaches have revealed valuable details about the interactions of nuclear structures such as the nuclear pore complex (NPC) [50,51] and nuclear lamina [52–56]. Moreover, BioID also has been used to screen for proteins in cell signaling pathways [57–59], tight junctions [60] and on chromatin [61]. Finally, protein interactions between viruses and their hosts contributing to the outcomes of viral infections have been studied by BioID. For example, interacting partners have been identified for the Gag protein of human immunodeficiency virus type 1 (HIV-1) [62–64], tegument protein UL103 of cytomegalovirus [65], latent membrane protein 1 of Epstein-Barr virus [66], and Zika virus-encoded proteins [67].

Here we used BioID approaches [50–53,55] combined with interactome-based mass spectrometry (MS) microscopy analysis [68] to investigate the nuclear interactions and nuclear localization of CPV NS2 both in noninfected and infected cells. The nuclear NS2 interactome identified by BioID included several components of different chromatin remodeling and DDR complexes. Furthermore, observations from assays with NS2 mutants suggested that N-terminal NS1/2-common splice donor mutant did not produce functional viral NS2 protein, resulting in inefficient chromatin remodeling regulation and DDR response. Altogether, our studies link CPV NS2 to novel functions in the nucleus and provide a platform for further functional analyses of NS2.

## Results

### Production of NS2 mRNA is temporally increased in infection and NS2 is localized into nucleoli

Since the expression of both CPV nonstructural proteins is controlled by the P4 promoter, we were interested in examining the expression levels of NS1 and NS2 mRNA at different times post infection. The quantitative reverse transcription PCR (RT-qPCR) using specific primers that distinguished the different transcripts showed that both NS1 and NS2 mRNAs were detectable at 4 hours post infection (hpi) in NLFK cells (Fig 1A). At 6 hpi, the levels of both mRNAs were higher compared to the control 18s rRNA levels, and both continued to increase until 24 hpi. The Student's t-test showed no statistically significant difference between the quantities of NS1 and NS2 mRNAs (p>0.05) at any time point. This data shows that NS2 is expressed early and throughout the viral replication. The early expression of NS2 is consistent with previous MVM findings [69], however, in contrast to MVM studies the expression level of CPV NS2 mRNA continued to increase during infection. MVM NS2 has a short half-life, which may account for the relatively small amount of nuclear NS2 detected in MVM infection [70].

To characterize potential NS2 functions during CPV replication, we next examined the association of NS2 with specific sites in the host cell nucleus. Confocal microscope images showed that NS2-EGFP expressed in noninfected HeLa cells was mostly localized near the rim of the nuclear periphery and in distinct nucleolar foci (Figs 1B and S1). A similar pattern of intranuclear localization was observed in CPV-infected cells, but the nucleolar distribution of NS2-EGFP was more diffuse than in noninfected cells. The BirA*-tagged NS2 used in BioID assays localized diffusely in noninfected cells. In infected cells, the BirA*-tagged NS2 distribution was mostly similar to noninfected cells, however, some local accumulation close to the nucleoli and viral replication center identified by NS1 was detected (Fig 1C). The nucleoli were identified by the exclusion of chromatin and NS1 labels. The homogenous nuclear distribution of the BirA*-tagged NS2 is consistent with NS2 distribution in infected NLFK cells identified by antibody against NS2 (S2 Fig). The BirA*-tagged NS2 was also localized in the cytoplasm both in noninfected and infected cells, however, the expression of the recombinant protein was decreased in infected cells. As a verification of nucleolar NS2 localization, we also found that in many wild type (wt) virus infected NLFK cells the antibody-stained NS2 colocalized with the nucleoli identified by nucleolin staining (Fig 1D). These results suggested that NS2 is often accumulated to nucleoli, where some of the essential chromatin remodeling and DDR factors reside [71–74].

### BioID links NS2 to chromatin modeling

We used BioID to identify binding partners for CPV NS2. This technique is based on the fusion of a biotin protein ligase (BirA) to the protein of interest, which leads to the biotinylation of proximal proteins of the fusion protein. BioID allows to probe both the stable, transient and proximal interactions of NS2 [52,53]. Our BioID data analysis of BirA*-tagged NS2 identified a total of 122 unique high-confidence interactions in transfected Flp-In T-REx 293 cells (Bayesian false discovery rate, BFDR, cutoff of <0.01 or <0.05), consisting of 44 interactions seen only in noninfected cells, 17 seen only in infected cells (24 hpi), and 61 seen in both (Fig 2A and S1 Table). The NS2-associated proteins in noninfected cells may represent interactions of NS2 that are independent of infection. Gene Ontology (GO) annotation analyses of biological processes, (Fig 2B and S2 Table), indicated the biotinylated proteins associated with NS2 involved in important cellular functions such as chromatin organization (GO:0006325, 34 proteins) and DDR (GO:0006974, 14 proteins) in both noninfected and infected cells.

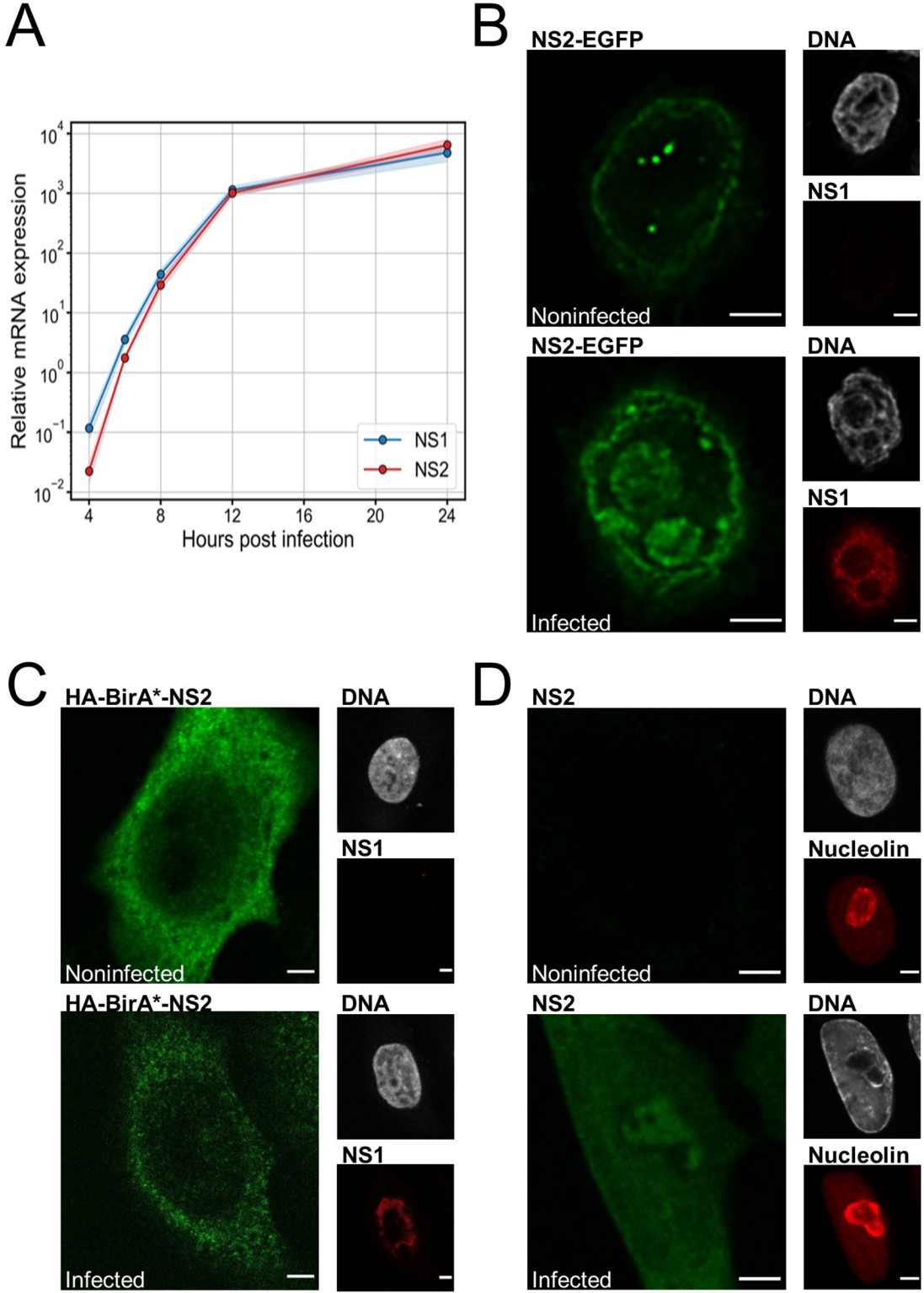

**Fig 1. CPV infection leads to simultaneous expression of NS1 and NS2 genes and nucleolar accumulation of NS2. (A)**
Relative expression levels of NS1 (blue) and NS2 (red) measured by RT-qPCR in infected NLFK cells between 4 and 24 hours post infection (hpi). The blue and red shadings around the lines indicate the standard error of the mean (SEM, n = 3). (**B**) Representative confocal images of HeLa cells transfected with NS2-EGFP (green) and (**C**) BirA*-tagged NS2 (green) at 24 hpt in noninfected and infected cells at 24 hpi. The localization of NS1 (red) is shown in cells with DAPI-stained nucleus (gray). (**D**) Representative confocal images of noninfected and infected NLFK cells at 24 hpi stained with antibodies against NS2 (green) and nucleolin (red). Gray corresponds to DAPI staining of DNA. Scale bars, 3 μm.

**A**

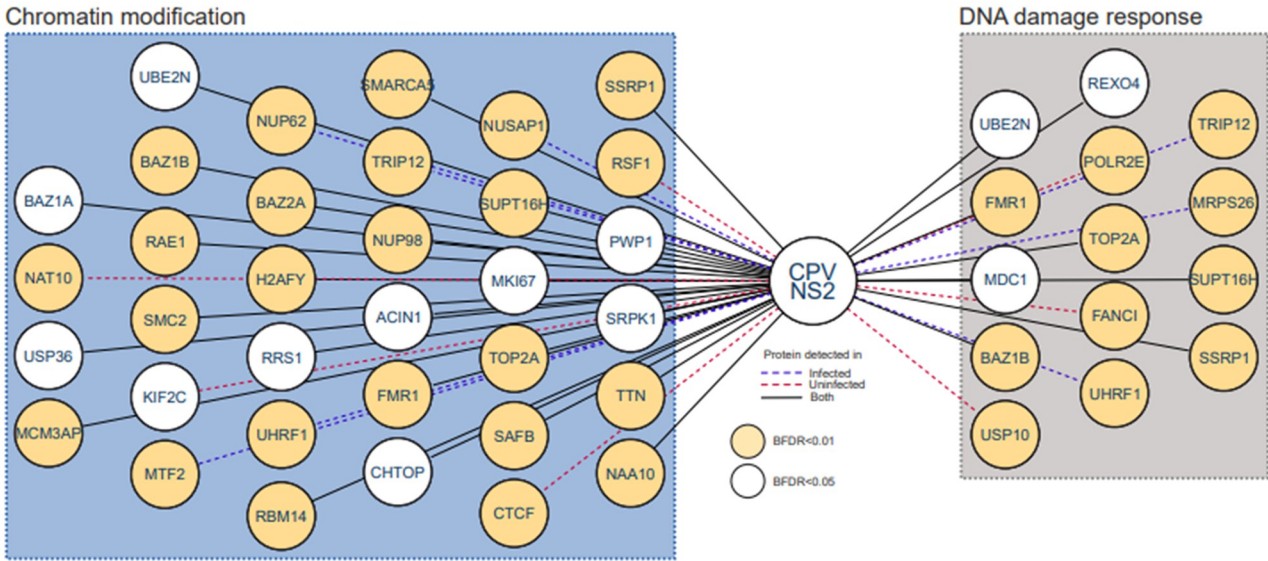

**B**

| GO Biological process | Count in infected | Count in noninfected | Total identified |
|---|---|---|---|
| Regulation of chromosome organization | 10 | 10 | 12 |
| Chromosome condensation | 5 | 4 | 5 |
| Chromosome organization | 25 | 24 | 28 |
| Regulation of chromatin organization | 8 | 7 | 9 |
| Negative regulation of chromosome organization | 5 | 4 | 6 |
| Nucleosome positioning | 1 | 3 | 3 |
| Nucleosome organization | 4 | 6 | 6 |
| Chromatin organization | 15 | 15 | 17 |
| Chromatin assembly or disassembly | 4 | 6 | 6 |
| Histone modification | 9 | 9 | 10 |
| Covalent chromatin modification | 9 | 9 | 10 |
| Cellular response to DNA damage stimulus | 11 | 11 | 14 |

**Fig 2. Nuclear interactors of NS2.** Identification of NS2 proximity interactome using BioID. (**A**) Schematic picture of 48 high-confidence BioID NS2 interactors related to chromatin modification and DNA damage response cellular processes. Interactors are presented by their gene names. Association of NS2 (HA-BirA*-NS2) is shown either in the presence (24 hpi, dark blue dashed line) or in the absence (red dashed line) of infection or in both cases (black solid line). (**B**) Table showing the GO functional annotation clusters for the NS2 BioID dataset. The functional annotation chart of NS2 high confidence interactors was created by PANTHER classification system for GO Biological process overrepresentation (http://www.pantherdb.org/) using the default BFDR < 0.01 (yellow circle) and < 0.05 (white circle) filters. The GO terms are grouped hierarchically by PANTHER to illustrate relations between functional classes and are colored based on relatedness to the two identified functional association groups: chromatin modification (blue) and DNA damage response (gray).

Since BioID suggested that NS2 associates with several nuclear proteins, the nuclear distribution of those NS2 interactors was further clarified using the interactome-based MS microscopy analysis. This method creates a high-precision reference molecular context proteome map generated by a combination of affinity purification mass spectroscopy and BioID. Three nuclear components–the nucleolus, chromatin and the nuclear envelope (NE)—were represented by the groups of cellular proteins proximal to rRNA 2′-O-methyltransferase fibrillarin, histone H3.1, and prelamin-A/C, respectively [68]. This proteome map analysis revealed that in noninfected cells the NS2 interactome was most similar to the NE marker, while sharing

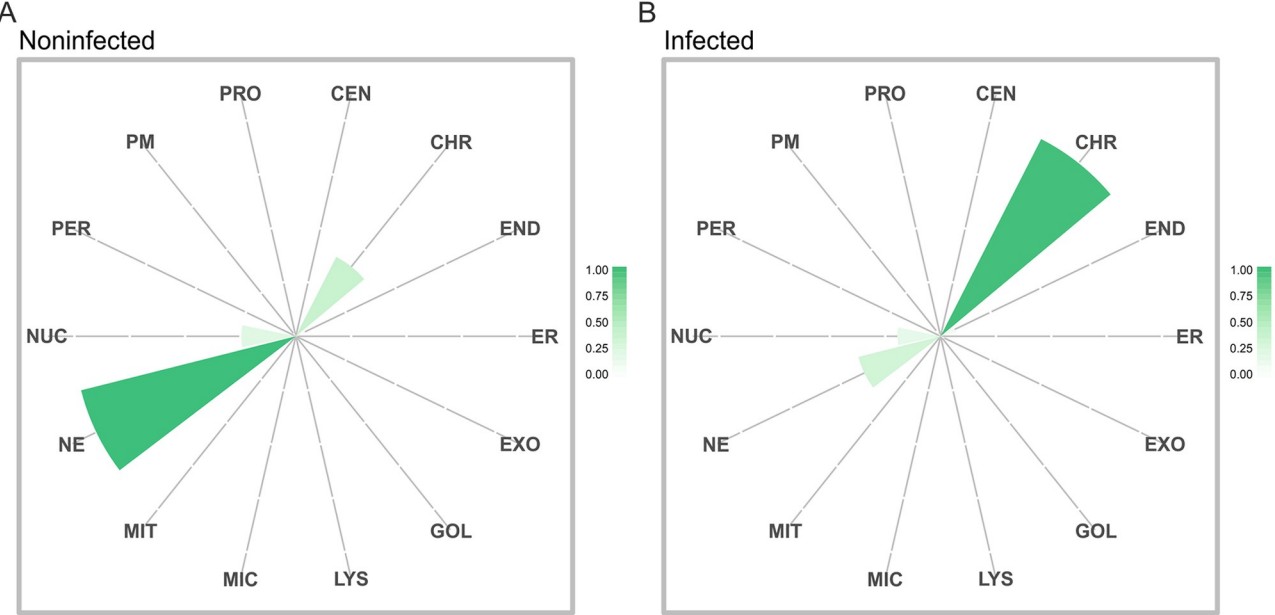

**Fig 3. Molecular context of NS2 varies in noninfected and infected cells.** The interactome-based localization of CPV NS2 was analyzed with mass spectrometry (MS) microscopy in (A) noninfected and (B) infected stably HA-BirA*-NS2 expressing Flp-In T-REx 293 cells. The polar plots show the location of NS2 protein associations based on cellular marker proteins. Each sector represents one subcellular location defined by our reference database. The color assigned to each of the localization is based on the annotation frequency (light green: 0–0.25; green: 0.25–0.5; dark green: 0.5–1). The markers used for nucleoli (NUC), chromatin (CHR) and nuclear envelope (NE) were rRNA 2′-O-methyltransferase fibrillarin, histone H3.1 and prelamin-A/C, respectively.

some interactors with nucleoli and chromatin (Fig 3A). Importantly, in infected cells the NS2 interactome shifted clearly towards chromatin (Fig 3B). This suggested that during infection NS2 is associated with proteins connected to the organization and modification of the host genomic DNA, and thereby further assessment of NS2 interactions was required.

## NS2 is associated with proteins linked to chromatin organization

The NS2 BioID interactome and GO annotation analyses suggested an association between the viral NS2 protein and cellular components involved in chromatin organization. BioID based interactome linked NS2 to four different complexes which belong to the major ATP-dependent chromatin remodeling complex family ISWI (Fig 4A and S1 and S2 Tables) [75]. The mammalian ISWI complexes identified were the nucleolar chromatin remodeling complex (NoRC), WSTF-ISWI chromatin remodeling complex (WICH), the ATP-utilizing chromatin assembly and remodeling factor complex (ACF), and the remodeling and spacing factor (RSF) complex.

One of the NS2 high-confidence (BFDR <0.01) BioID hits was nucleosome-remodeling helicase matrix-associated actin-dependent regulator of chromatin A5 (gene name *SMARCA5*, also known as SNF2H) [76,77] which is associated with all four ISWI complexes. NS2-associated proteins included bromodomain adjacent to zinc finger domain 2A (BAZ2A, also known as TIP5, BFDR <0.01) [71,72], a regulator of SMARCA5 in NoRC. [71,72]. Notably, the highest average spectral counts in our BioID analysis were produced by the proliferation marker protein Ki-67 (MKI67; BFDR 0.02 in infected and 0.04 in noninfected cells). Ki-67, a nuclear protein expressed in actively dividing mammalian cells, is involved in the organization of heterochromatin and it also interacts with BAZ2A [78–80]. Moreover, NS2 was associated

## A Chromatin remodelling

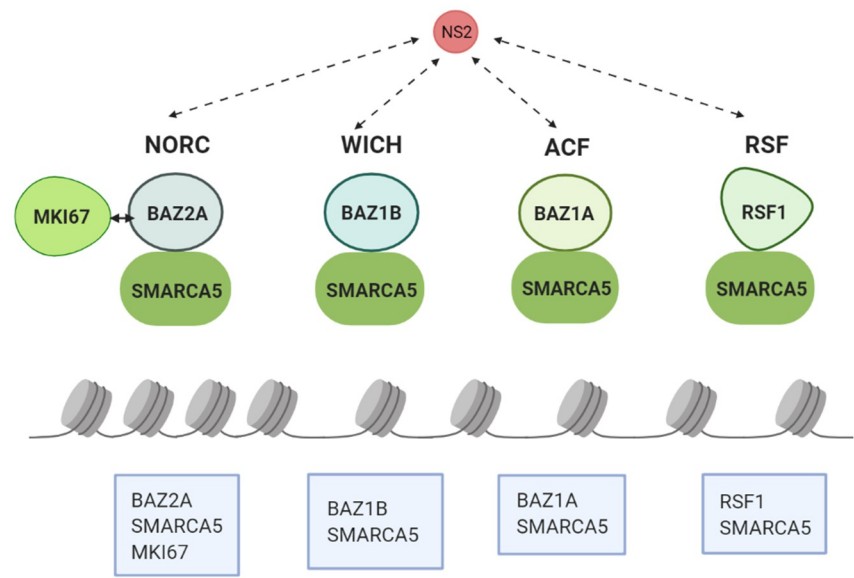

## B DNA damage response

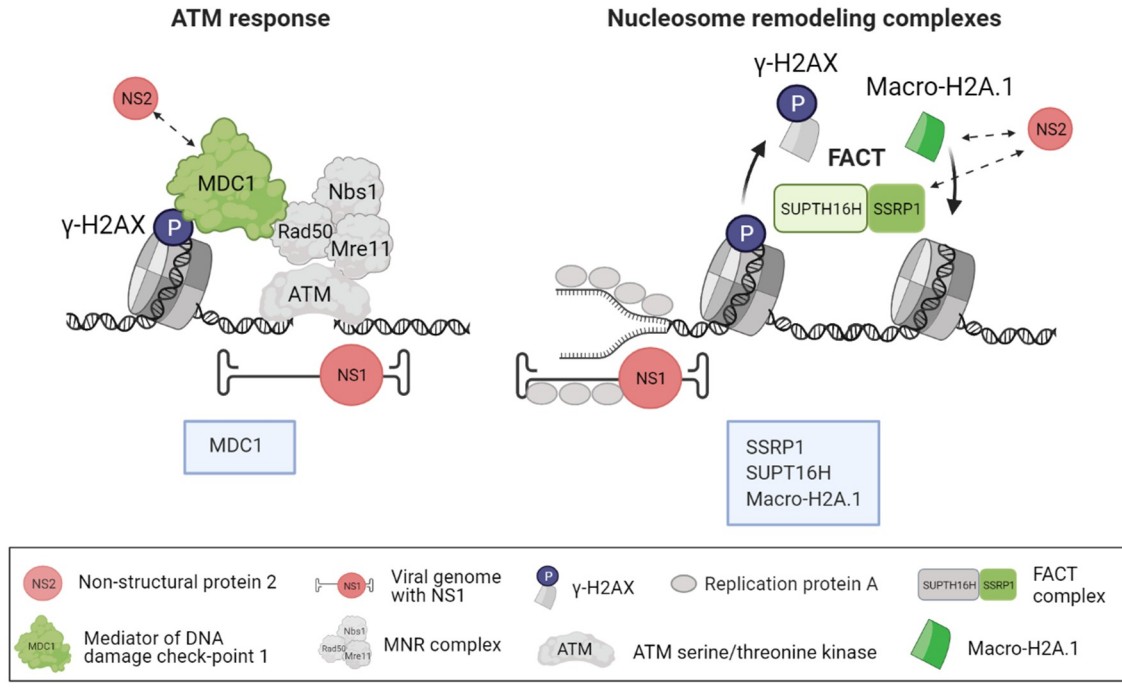

**Fig 4. NS2-associated proteins play important roles in chromatin remodeling and DDR machineries.** The CPV NS2 protein is linked with certain key factors that control chromatin remodeling and DDR pathways to optimize cellular conditions for viral replication. This schematic picture shows the major associations of NS2 (red) with cellular proteins (shades of green). Dashed arrows represent key factors of chromatin binding and DDR machineries interacting with NS2. (**A**) The cellular chromatin modification machinery includes chromatin remodeling complexes such as NORC, WICH, ACF and RSF. (**B**) DNA damage response (DDR) factors contain major upstream mediators of ATM response such as DNA damage checkpoint 1 (MDC1) protein and proteins

included in chromatin and nucleosome remodeling complexes. MDC1 interacts with phosphorylated γ-H2AX and mediates the recruitment of DDR response proteins to the damage site. The DDR downstream FACT complex functions as a nucleosome remodeler facilitating transcription. During replication stress, FACT orchestrates the replacement of γ-H2AX with macro-H2A.1 to the damage site. The location of parvoviral genomes and the viral replication protein NS1 (red) adjacent to DNA damage sites of the cellular chromatin are shown. Figure was created with BioRender.com.

with tyrosine-protein kinase BAZ1B (also known as the Williams syndrome transcription factor, WSTF, BFDR <0.01) [81] a component of WICH with SMARCA5. NS2 was associated with both BAZ1A (BAZ1A, also known as ATP-utilizing chromatin assembly and remodeling factor 1, ACF1; BFDR <0.05) and remodeling and spacing factor 1 (RSF1, BFDR <0.01). ACF [81] and RSF [82,83,84,85] are formed when SMARCA5 combines either with BAZ1A or with RSF1. The potential role of NS2 in the regulation of transcription was further supported by its interaction with a member of the histone H2A family, core histone macro-H2A.1 (H2AFY, BFDR <0.01), which has been shown to associate with transcription repression [86]. The association between NS2 and SMARCA5, BAZ1A, BAZ1B, BAZ2A, Ki-67, and macro-H2A.1 were detected both in noninfected and infected cells. Together, these data indicate that CPV NS2 associates with proteins of four chromatin-modifying complexes. Therefore, it is possible that NS2 is involved in the manipulation of chromatin modeling processes potentially inducing modifications of both cellular and viral DNA.

## NS2 is linked to DDR-associated proteins

We next sought to confirm the association of NS2 with DDR factors necessary for parvoviral replication [34,87]. The progression of cell cycle is coordinated by DNA damage checkpoints, which delay or stop the cell cycle before or during DNA replication in the presence of damaged DNA [88,89]. DDR includes complex signaling cascades that require the actions of various proteins that function as DNA damage sensors, transducers, mediators, and effectors. One of the upstream mediators in DDR is DNA damage checkpoint 1 (MDC1, BFDR <0.05) protein. The BioID analysis suggested NS2 interactions with MDC1 in both noninfected and infected cells, as well as associations with DDR downstream factors, including components of the nucleolar facilitator of chromatin transcription (FACT) complex and WSTF-including nucleosome remodeling complex (WINAC) (Fig 4B and S1 and S2 Tables). FACT acts as a critical chaperon for histones in nucleosome reorganization during replication, and in the detection and response of DNA damage [90,91], stabilizing chromatin as a whole by suppressing cryptic transcription [92,93]. FACT complex subunits, structure specific recognition proteins (SSRP1, BFDR <0.01) and SUPT16H (SPT16, BFDR <0.01), were identified as high-confidence NS2 interactors both in noninfected and infected cells. SSRP1 is a histone chaperon involved in transcriptional regulation, DNA replication and damage repair [94–97]. SUPT16H also functions independently of FACT when it forms WINAC with BAZ1B (BFDR <0.01) [98,99]. WINAC is an ATP-dependent chromatin remodeling complex, which is associated with a variety of DNA processing functions. Furthermore, FACT is linked to the activation of p53, a central tumor suppressor, the stability of which is further regulated by ubiquitin carboxyl-terminal hydrolase 10 (USP10, BFDR <0.01), also identified as an NS2 interactor in BioID of noninfected cells. USP10 relocates to the nucleus in response to DDR and promotes the deubiquitination of p53 [100]. Similar function is served by E3 ubiquitin-protein ligase (TRIP12, BFDR <0.01), an NS2 interactor in infected cells, which indirectly regulates p53 activity by affecting its ubiquitination. In addition to the previously mentioned and other ubiquitination-related enzymes in the BioID results, ubiquitin-conjugating enzyme E2 N (UBE2N, BFDR <0.05) was also identified in the absence and presence of infection. It may act in non-

degradation ubiquitination targeting and DNA damage repair [101]. Taken together, these results demonstrated that NS2 is associated with the DDR signaling proteins during infection. Since DDR has a clear potential role in parvovirus replication [34,87], NS2 may recruit DDR effector proteins to regulate viral replication.

## Proximity-dependent interaction analysis corroborates NS2 BioID findings

To confirm the BioID findings of NS2 association with nuclear proteins, the cellular distributions of four NS2-associated proteins from infected and noninfected cells, SMARCA5, MDC1, macro-H2A.1 and SSRP1, which represent chromatin modeling and DDR functions, were studied in noninfected HeLa cells expressing NS2-EGFP and in infected NLFK cells. As described above, NS2-EGFP was mostly localized to the NE and into nucleolar foci (Figs 1B and S1) whereas antibody-labeled NS2 showed more homogenous nuclear and nucleolar distribution (Figs 1D and S2). SMARCA5 and MDC1 and macro-H2A.1 were distributed throughout the nucleus in transfected cells, whereas macro-H2A.1 and γ-H2AX localized to the nuclear edges in infected cells. SSRP1 was mostly located into specific nucleoplasmic foci in both transfected and infected cells (S1 and S2 Figs). Confocal imaging of infected HeLa cells at 24 hpi showed that SMARCA5 and SSRP1 occupied sites that overlapped with the NS1-containing replication centers. The transcription repressor macro-H2A.1 was not found in the replication center area, but was localized close to the replication centers in regions without NS1 (S3 Fig). Together, confocal imaging therefore suggested that SMARCA5, MDC1, SSRP1, and macro-H2A.1 associated or localized adjacent to regions containing either NS2 or NS1.

To further asses NS2 interactions and to move beyond diffraction-limited observations of protein localizations, protein-protein interactions were studied with Proximity Ligation Assay (PLA) which is an immunodetection technique that generates a fluorescent signal only when two antigens of interest are within 40 nm of each other [102]. All the tested NS2 interactors, SMARCA5, SSRP1, and macro-H2A.1, gave punctate nuclear PLA signals which localized often close to the nucleoli in cells transfected with NS2-EGFP. Here, anti-GFP was used to form a PLA probe pair with antibodies against SMARCA5, SSRP1, or macro-H2A.1 (Fig 5A). The negative viral capsid antibody control in non-infected cells and technical probe controls in NS2-EGFP transfected cells (Figs 5B and S4A and S4B) indicated that the background in the nuclear area was low (approximately 0.9 ± 0.3 and 0.7 ± 0.2 PLA foci/nucleus, respectively, n = 15). A quantitative analysis of 3D data showed that within the segmented nuclei (S4C Fig), the highest number of PLA foci was detected for NS2 with macro-H2A.1 (135 ± 12 PLA foci/nucleus). A smaller number of PLA foci was detected for NS2 with SMARCA5 and SSRP1 (55 ± 11 and 7.5 ± 1.4 PLA foci/nucleus, respectively) (Fig 5C). The Poisson regression model [103] showed that NS2 with SMARCA5, macro-H2A.1 and SSRP1 had a significantly higher (p<0.01) number of nuclear PLA foci compared to the negative control. Moreover, NS2 interactions with SMARCA5, MDC1, macro-H2A.1, and SSRP1 were further confirmed using PLA in infected cells. Assay was performed by probing anti-NS2 antibody together with antibodies against BioID identified interactors. Similar to the NS2-EGFP transfected cells, representative examples shown in Fig 6A demonstrated proximal localization of PLA signals and the nucleoli. The highest number of PLA foci in comparison to negative control (1.9 ± 0.5 PLA foci/nucleus, Fig 6B) was detected for macro-H2A.1 (40 ± 16 PLA foci/nucleus) followed by MDC1 (22 ± 6 PLA foci/nucleus), SMARCA5 and SSRP1 foci (6.5 ± 1.5 and 3.7 ± 0.8 PLA foci/nucleus, respectively) (Fig 6C). The technical controls indicated that the background in the nuclear area was low (approximately 0.3 ± 0.3 and 1.1 ± 0.9 PLA foci/nucleus, respectively, n = 15). Together, our interaction analyses of prominent NS2 interaction partners support BioID

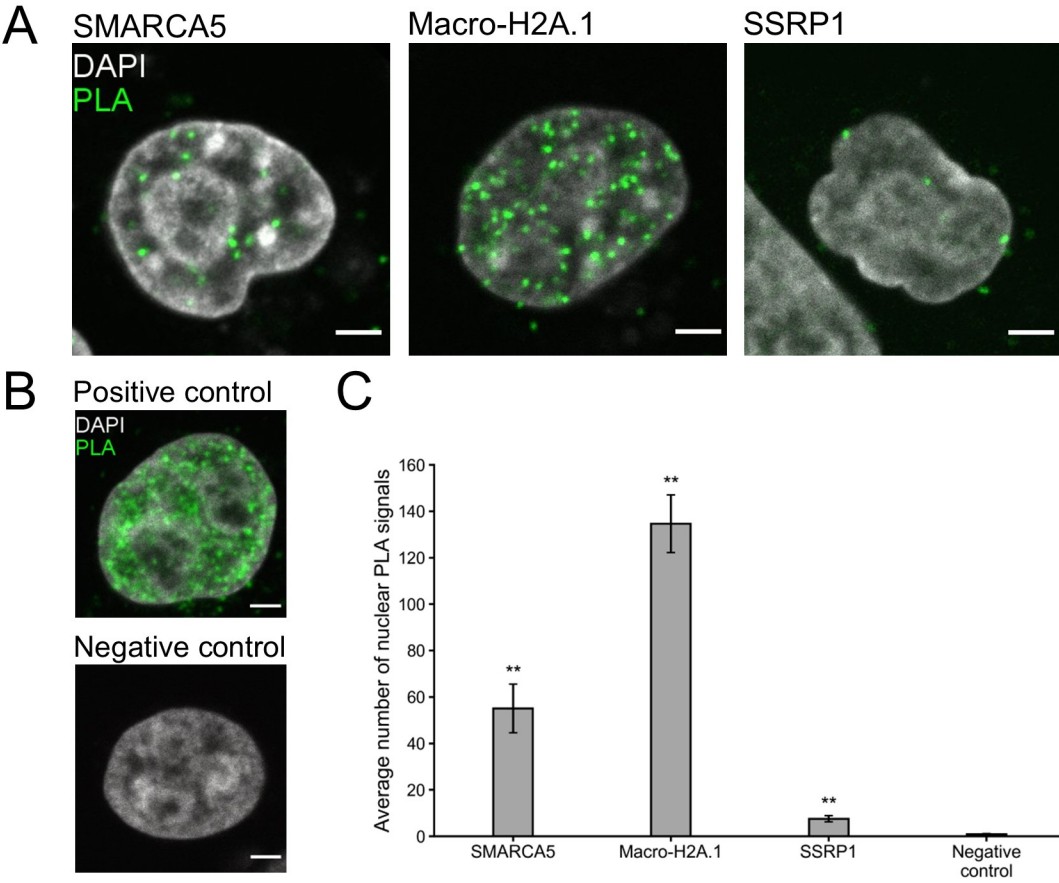

**Fig 5. Close proximity analysis verifies the association of NS2 with key BioID interactors.** Proximity Ligation Assay (PLA) assay was performed in HeLa cells transfected with NS2-EGFP and representative cellular proteins identified by BioID. The proteins were chromatin remodeling factor (SMARCA5), transcriptional repressor (macro-H2A.1), and histone chaperon of the FACT complex (SSRP1). (**A**) Representative single confocal cross-sections showing the localization of PLA signals (green) and the nuclei stained with DAPI (gray). (**B**) Positive and negative controls include PLA with antibodies against VP2 capsid protein and intact capsids in infected cells at 20 hpi and in noninfected cells. Scale bars, 3 μm. (**C**) The graph shows the average number of PLA signal foci per nucleus (n = 15) analyzed from 3D reconstructions of confocal images. The error bars show the standard error of the mean (SEM), and the samples having a significantly higher number of nuclear PLA foci compared to the negative control are denoted as ** (p<0.01). The significance was assessed using Poisson regression model.

results demonstrating that NS2 is likely associated with proteins of DNA modeling and DDR pathways.

## Mutation of NS2 leads to changes in chromatin remodeling

Our BioID analysis showed that NS2 is associated with cellular proteins involved in chromatin organization. To further validate our findings, we compared NS2 mutant clones to the wild type (wt) infectious clone to analyze their effect on chromatin organization. In these assays, cells were transfected either with the wt infectious clone or with NS2 splice donor or splice acceptor mutants (G533A and A2003T, respectively) which were designed to disrupt the expression of NS2 but not to affect the amino acid sequence of NS1 [6]. These previously described mutants are based on the MVM mutants that have been shown to alter the expression of MVM NS2 and lead to abortive infection [12,13,104]. CPV NS2 mutants have been created by introducing termination codons into the NS2 coding sequence of the infectious clone to a site upstream of the common NS1/2 splice donor or downstream of the NS2-specific splice

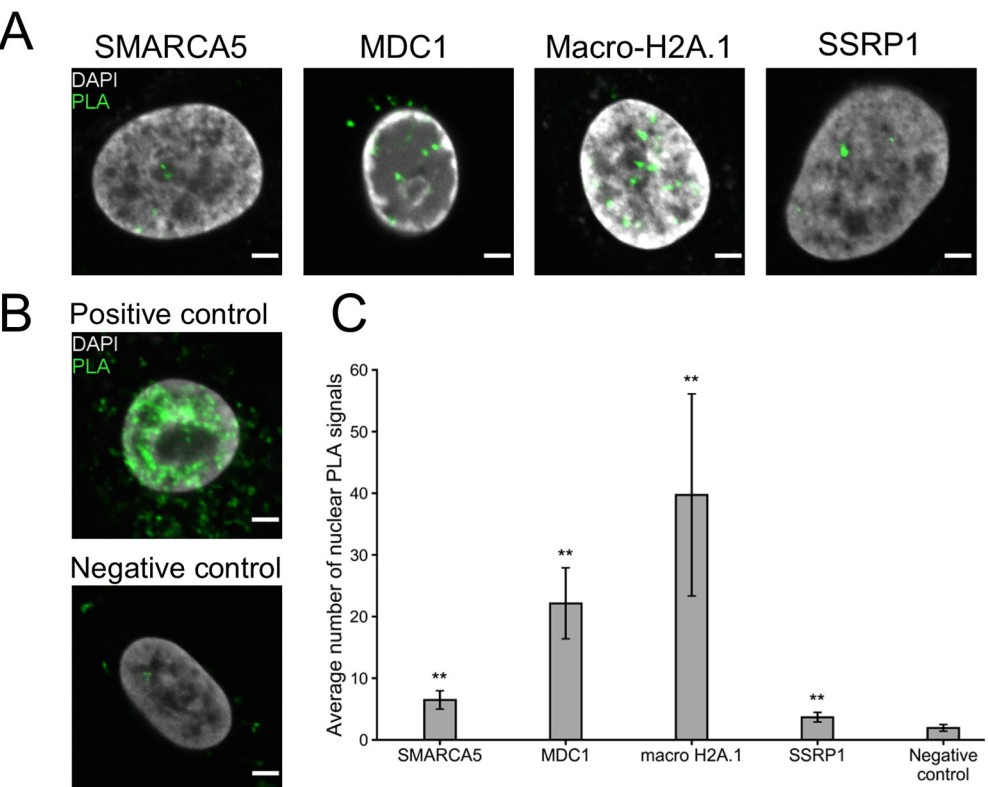

**Fig 6. NS2 interacts with host proteins identified by BioID in infected cells.** Analyses of NS2 interaction with SMARCA5, MDC1, macro-H2A.1, and SSRP1 in infected NLFK cells at 24 hpi. (**A**) Representative single confocal cross-sections of PLA signals (green) in the DAPI-stained nuclei (gray). (**B**) Images of positive and negative PLA controls prepared by using antibodies against VP2 and intact capsids at 24 hpi and in noninfected cells. Scale bars, 3 μm. (**C**) The analyses of 3D reconstruction of confocal PLA data showing the average number of PLA signal foci/nucleus (n = 15). The error bars show the standard error of the mean (SEM), and the samples with a significantly higher number of nuclear PLA foci compared to the negative control are denoted as ** (p<0.01). The significance was assessed using Poisson regression model.

acceptor [6]. The original publication demonstrated that the production of viral capsid proteins (VP1/VP2) and viral DNA were all significantly decreased in cells transfected with the NS2 splice donor mutant virus, while transfection with the splice acceptor mutant was comparable with wt CPV. Characterization of mutants by immunoprecipitation analyses (antibodies against the NS2 C-terminus) showed that both mutants were unable to produce intact NS2 protein [6]. However, the interpretation of mutant-induced effect on viral life cycle was complicated by RT-PCR characterization which revealed that alternative sequences used to splice the message RNA were present in both mutants [6]. It should be also noted that, that the N-terminal end of NS1, which share a common N-terminal domain with NS2, might be produced in some circumstances, and might play a role in virus-induced chromatin remodelling, although that remains to be defined.

Here, we used confocal microscopy intensity analysis to study the distribution pattern and relative amount of nuclear DNA (stained with DAPI), euchromatin (labeled for H3K27ac), and heterochromatin (labeled for H3K9me3) in HeLa cells transfected with the wt CPV clone and NS2 mutants at 24 hours post transfection (hpt). Comparison of wt and mutant viral transfection revealed that the donor NS2 mutant clone displayed a clearly higher total intensity of the euchromatin (Fig 7A and 7B). It is important to note that the increased amount of

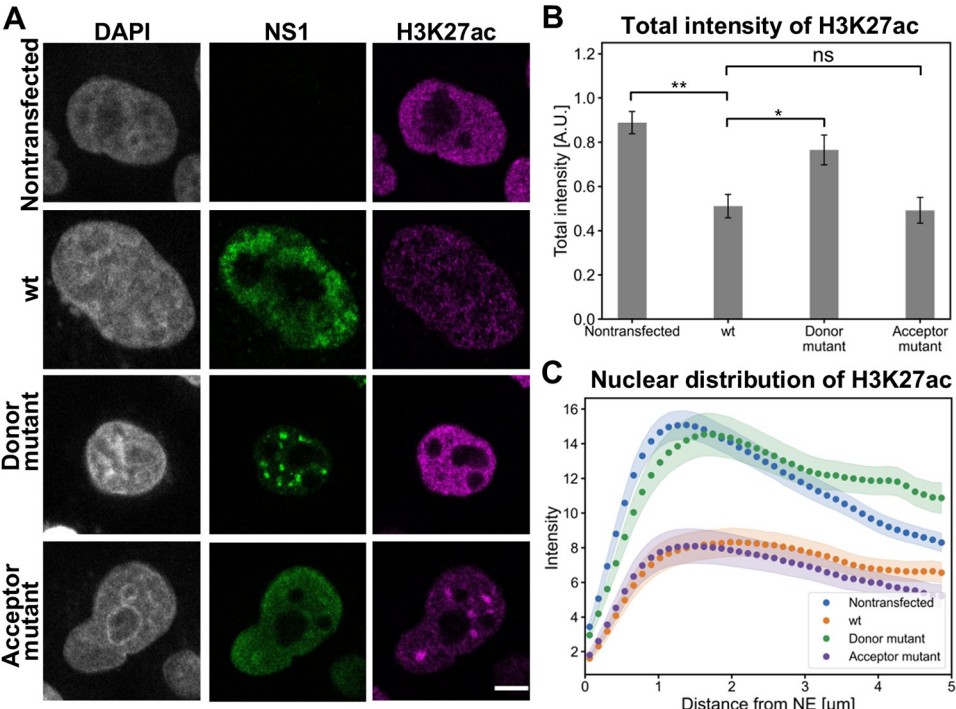

**Fig 7. NS2 mutation induce changes in the amount and distribution of euchromatin.** (**A**) Representative confocal images show the nuclear localization of euchromatin marker H3K27ac (magenta), NS1 (green), and DAPI staining (gray) in nontransfected HeLa cells, and cells transfected with wt, NS2 donor and acceptor mutants at 24 hpt. (**B**) Total fluorescence intensities of euchromatin. (**C**) Nuclear distribution of H3K27ac as a function of increasing distance from the NE in nontransfected, wt and NS2 mutants transfected cells (n = 29). The error bars show the standard error of the mean. Statistical significances were determined using Dunnett's multiple comparison test. The significance values shown are denoted as ** (p<0.01), * (p<0.05) or ns (not significant). Scale bars, 5 μm.

euchromatin in splice donor mutant transfected cells was accompanied by significantly lower total intensity of DAPI staining (S5A Fig). This indicates that infection affects the epigenetic regulation of host chromatin, and the splice donor mutation of NS2 results in a reversion towards the decondensation state seen in noninfected cells. Distribution analyses of both nuclear DNA (S5B Fig) and euchromatin (Fig 7C) as a function of the distance from the NE further demonstrated their localization both in the nuclear periphery and in the central region of the nucleus. Additionally, transfection by all clones resulted accumulation of heterochromatin at the perinuclear region and around the nucleoli (S6A Fig). Notably, NS2 mutations did not appear to affect the relative intensity of the heterochromatin label (S6B Fig). Distribution analyses of heterochromatin further confirmed its localization both close to the nuclear periphery and in the center of the nucleus (S6C Fig). Altogether, our findings demonstrated clear changes in the quantity of euchromatin in response to NS2 donor mutant transfection, suggesting that N-terminal sequence of NS2 could have a currently undefined role in chromatin remodeling during infection. This supports our BioID results demonstrating that NS2 is associated with proteins linked to chromatin organization.

## The level and localization of DDR proteins change in the presence of NS2 mutant

Cellular DNA double-strand break repair sites recruit several upstream DDR proteins such as γ-H2AX [105–107] and MDC1 [108] (Fig 4B). DNA damage proteins have also been

previously observed to localize to the periphery of MVM viral replication centers [109]. As shown by our BioID data, CPV NS2 is associated with MDC1 (Fig 2 and S3 Table). To further examine the role of NS2 interactions in DNA damage, we analyzed the localization and intensity of γ-H2AX and MDC1 and their connection to replication centers in HeLa cells transfected with the wt CPV, splice donor and splice acceptor mutants at 24 hpt. Our studies demonstrated that both γ-H2AX and MDC1 localized next to the replication centers and accumulated close to the nucleoli and NE in wt or the splice acceptor mutant transfected cells, in contrast to relatively diffuse nuclear localization in nontransfected cells and in cells transfected with the splice donor mutant. NS1 accumulated in distinct nuclear foci in cells transfected with the splice donor mutant (Fig 8A and 8B). As expected, both γ-H2AX and MDC1 colocalized with DNA marker. Quantitative image analysis (Fig 8C) showed that wt and the splice acceptor mutant transfections led to relatively similar amounts of nuclear γ-H2AX, whereas it was significantly decreased in nontransfected cells and in the splice donor mutant transfected cells. The distribution analyses indicated that γ-H2AX was distributed close to the NE and in the central region of the nucleus in cells transfected with the wt and the splice acceptor mutant, whereas γ-H2AX was slightly more concentrated close to the NE in nontransfected cells and in the splice donor mutant transfected cells (Fig 8D). Intensity line profiles measured through the nucleoli verified close association between γ-H2AX and replication centers in close proximity to the nucleoli both in cells transfected with the wt and the splice acceptor mutant. The close localization between replication centers and H2AX was visible to a lesser extent in the cell transfected with the splice donor mutant (Fig 8E). Similar to γ-H2AX, the total intensity of MDC1 was clearly decreased in cells transfected with the splice donor mutant in comparison to cells transfected with wt or the splice acceptor mutant (Fig 8F), and it was localized slightly more toward the NE in the splice donor mutant transfected cells (Fig 8G). The line profiles of wt and the splice acceptor mutant transfected cells showed that replication centers and MDC1 were located close to each other and they both were located near the nucleoli (Fig 8H). The localization of MDC1 in the periphery of NS1-labeled replication center foci was indistinct. Taken together, our data reveals that transfection with the NS2 splice donor mutant resulted in significant changes in the amounts and distributions of DDR proteins. The decreasing association of H2AX and MDC1 with the viral replication centers in cells transfected with the splice donor mutant (Fig 8E and 8H) is interesting as MVM studies have demonstrated a close localization between replication centers and DDR proteins [34]. These findings show that N-terminal sequence of NS2 is required for the normal progression of DNA damage response in infection, and the mutation of splice donor site has an effect on the recruitment of H2AX and MDC1 to the replication region where they most likely associate with NS1 in wt infection.

## Mutation of NS2 leads to changes in the establishment of replication centers

MVM replication centers are established at cellular DNA damage sites, which provide necessary factors for the initiation of viral replication. The progress of infection requires association with new DNA damage sites, allowing the amplification of viral replication [109]. Therefore, as our experiments demonstrated that the mutation of NS2 splice donor site leads to a decrease in DDR proteins, we assessed how NS2 mutations affect the development of viral replication centers.

As shown above both wt and the splice acceptor mutant transfections induced the development of large replication centers, which filled most of the nuclear region at 24 hpi. In contrast, multiple small replication centers were formed in cells transfected with the splice donor mutant (Figs 7A, 7B, 8A and 8B). Confocal images showed that in wt and the splice acceptor

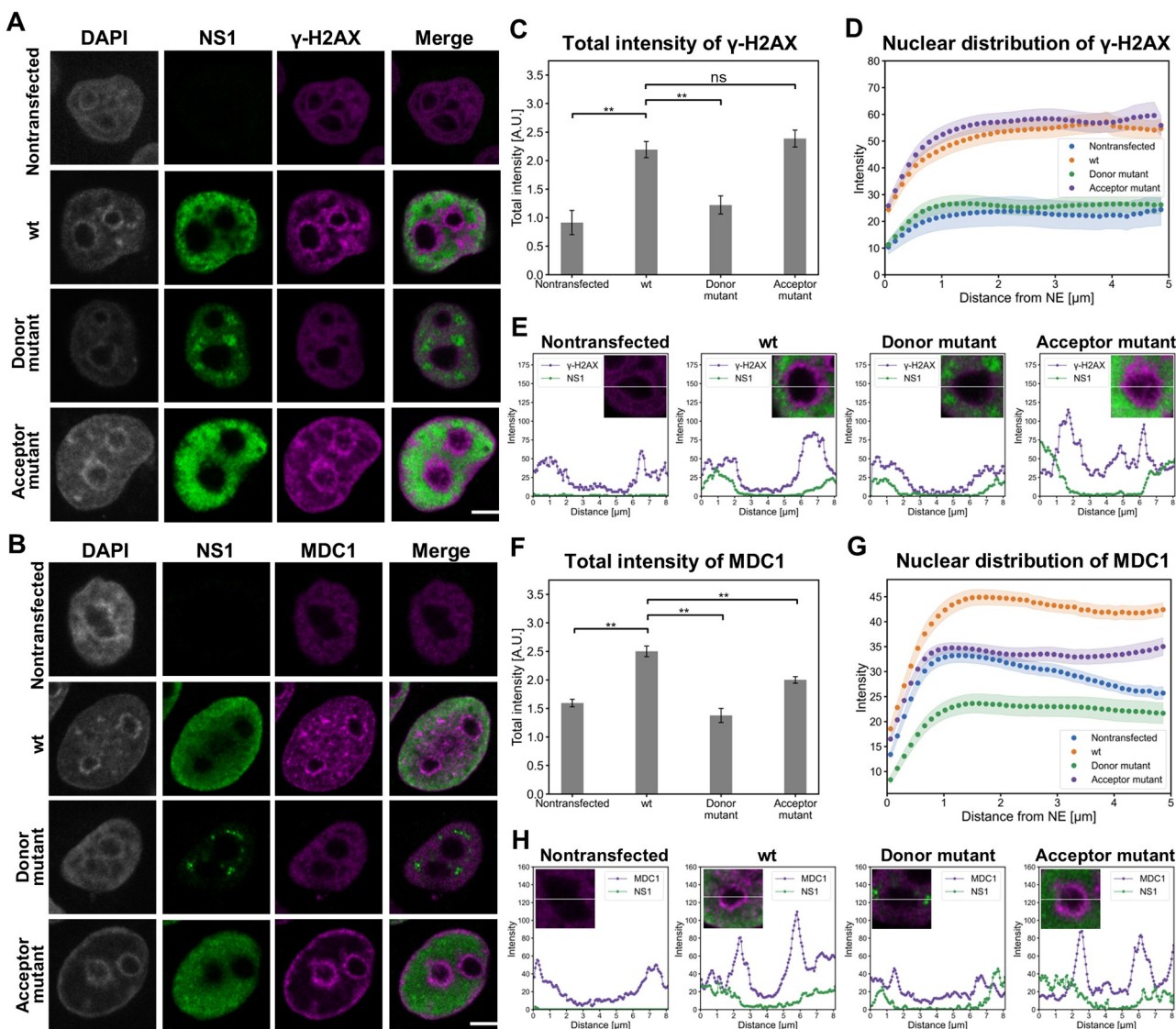

**Fig 8. Mutation of NS2 induce alteration in the amount and distribution of DDR proteins.** Representative confocal images show the nuclear localization of NS1 (green), (**A**) γ-H2AX (magenta), (**B**) MDC1 (magenta) and DAPI staining (gray) in HeLa cells without transfection or at 24 hpt with wt, NS2 splice donor and acceptor mutants. Scale bars, 5 μm. (**C**) Total fluorescence intensities of γ-H2AX together with (**D**) its nuclear distribution as a function of increasing distance from the NE in nontransfected, wt and NS2 mutants transfected cells (n = 29). (E) Intensity line profiles of γ-H2AX (purple) and NS1 (green) measured through the nucleoli. (**F**) Fluorescent intensities and (**G**) nuclear localization of MDC1 from the NE (n = 28). The error bars show the standard error of the mean. (H) Intensity line profiles of MDC1 (purple) and NS1 (green) measured through the nucleoli. Statistical significances were determined using Dunnett's multiple comparison test. The significance values shown are denoted as ** (p<0.01), * (p<0.05) or ns (not significant). Fluorescent intensity profiles of (**G**) γ-H2AX (magenta) with NS1 (green), and (**H**) MDC1 (magenta) with NS1 (green) in zoomed areas.

mutant transfections NS1 was seemingly localized to the nuclear periphery and excluded from the center of the nucleus where nucleoli are present. The NS1 foci were often localized around the nucleoli in the splice donor mutant transfected cells (Fig 9A). Comparison of NS1 intensities in transfected cells showed a significantly decreased amount of NS1 in the cells transfected with the NS2 donor mutant (Fig 9B). This data suggests that replication centers in the cells transfected with the splice donor mutant were initiated but failed to expand. This is consistent with the finding that most of NS1 in the cells transfected with the splice donor mutant was

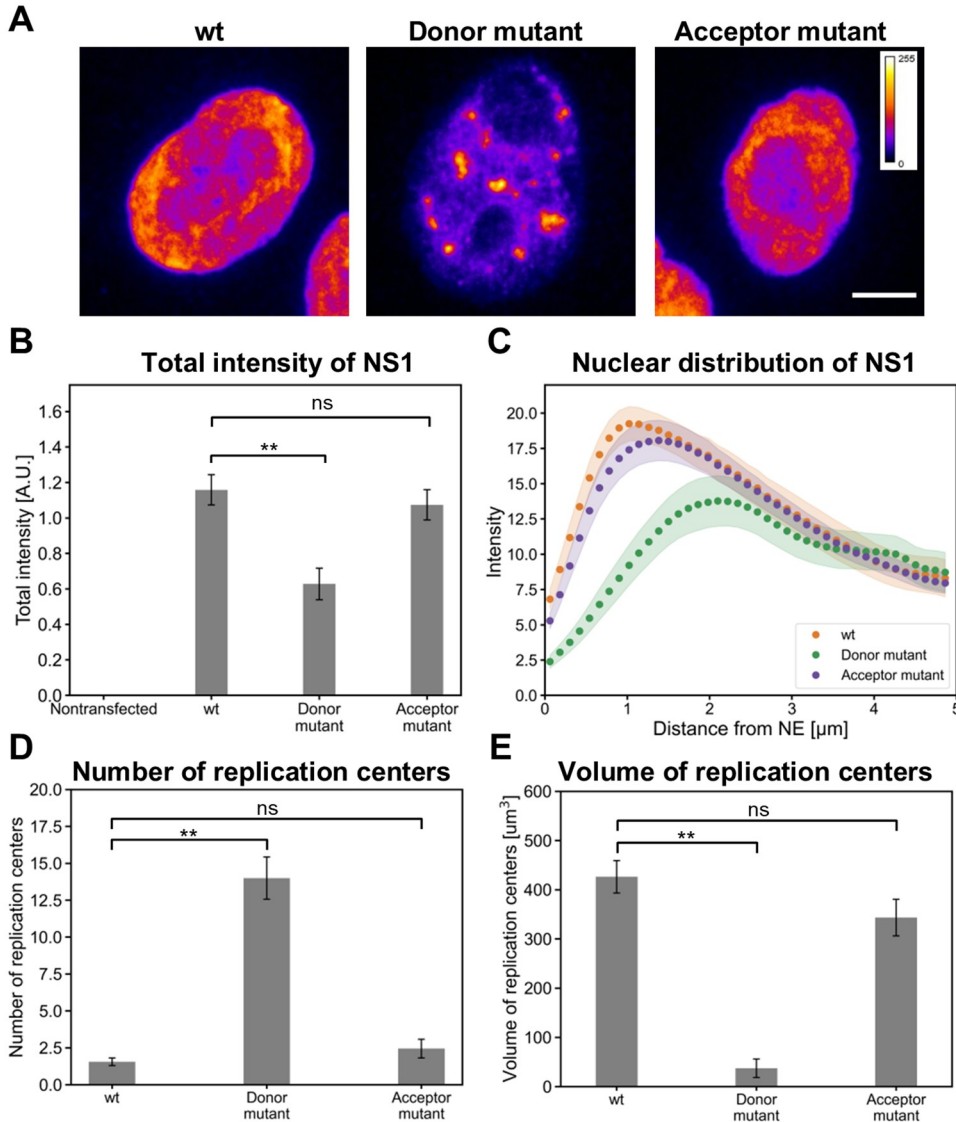

**Fig 9. NS2 mutation induce changes in the number and volume of replication centers.** (**A**) Representative pseudocolored sum projections of confocal image slices show the amount and distribution of nuclear NS1 in nontransfected, wt, splice donor and acceptor mutants transfected HeLa cells at 24 hpt. Calibration bar for pseudo coloring is shown. Scale bars, 5 μm. (**B**) Total fluorescence intensities of NS1. The error bars show the standard error of the mean. (**C**) Nuclear distribution of NS1 as a function of increasing distance from the NE in wt and NS2 mutants transfected cells (n = 29). (**D**) Numbers and (**E**) volumes of replication centers, identified by NS1 staining were analyzed in wt, splice donor and acceptor mutants transfected HeLa cells at 24 hpt (n = 29). The error bars show the standard error of the mean. Statistical significances were determined using Dunnett's multiple comparison test. The significance values shown are denoted as ** (p<0.01), * (p<0.05) or ns (not significant).

localized towards the center of the nucleus, while in the cells transfected with wt and the splice acceptor mutant NS1 was mostly found in the nuclear periphery (Fig 9C). The average number of replication center foci detected in cells transfected with the splice donor mutant was 14.0 ± 1.4 (n = 29), whereas in the cells transfected with wt and the splice acceptor mutant it was 1.6 ± 0.3 and 2.4 ± 0.6 (n = 29) (Fig 9D). This is consistent with the notion that the size of the NS2 splice donor mutant induced foci was small (40 ± 20 μm$^3$, n = 29) in comparison to

replication centers in the cells transfected with wt ($430 \pm 30$ µm$^3$, n = 29) and the splice acceptor mutant ($340 \pm 40$ µm$^3$, n = 29) (Fig 9E). As described earlier, parvoviral replication centers are associated with sites of cellular DNA damage, and progression of infection is connected to increasing DNA damage [14,34]. Our observation that mutation in N-terminal end of NS2 leads to a deficient development of viral replication centers supports a model that the role of NS2 in DDR is essential for the successful infection.

## Discussion

When viruses infect cell nuclei for their own benefit, some of the host structures and functions are left unchanged while others go through dramatic remodeling processes and functional changes. Parvoviruses possess a small genome and express only a limited number of proteins, and therefore rely heavily on the nuclear machinery present in the cellular S phase for their efficient replication. As the infection proceeds, extensive changes in the nucleus occur, including altered chromatin architecture and redistribution of DDR components [43,44,110]. As with other parvoviruses, the CPV NS1 performs several functions essential for viral replication. However, the role of NS2 has remained undefined. Our BioID results demonstrate that CPV NS2 protein associates with many cellular proteins involved in essential nuclear processes such as chromatin modification and DDR, and therefore has a potentially important role in chromatin remodeling and in interactions required for viral replication.

BioID analyses linked NS2 to proteins from four chromatin remodeling complexes of the ISWI family involved in nucleosome organization during nucleosome remodeling for the regulation of gene expression [111–113] (Fig 4A). The interaction of CPV NS2 with these complexes is not unexpected as dsDNA viruses and most likely parvoviruses are challenged by cellular defense mechanisms seeking to silence viral gene expression by chromatinization of the viral genome [25,31–33]. Also, another remodeling mechanism appears to facilitate transcription from the viral genome, as the two NS1 binding promoters of CPV, P4 and P38, contain histone modifications characteristic of an accessible euchromatin state [33].

Alternatively, viruses can induce changes in host transcription by epigenetic modifications, redirecting cellular resources for viral gene expression, or interfering with the immune responses of the host cell. Our results show that CPV NS2 is linked with proteins of the nucleolar NoRC complex. This complex is involved in downregulation of ribosomal gene transcription [71,72,114]. Cellular ribosomal proteins are essential for viral translation and replication, and herpes virus studies have shown that their expression increases during infection [115,116]. For human cytomegalovirus, restricting ribosome biogenesis by NoRC-induced silencing of rRNA expression reduces the progression of infection, and ribosome biogenesis may also regulate innate immune responses of the host [117]. The connection between NS2 and NoRC proteins may benefit CPV infection by ensuring ribosome sufficiency for viral protein synthesis. The NS2 interaction with NoRC may also be associated with regulation of host innate responses—however, this requires further investigation. NS2 is also associated with WICH chromatin remodeling complex proteins involved in RNA polymerase and ribosomal RNA transcription [118]. These results suggest that NS2 might cooperate with WICH proteins to facilitate transcription on host chromatin, thereby contributing to the enhancement of RNA polymerase and ribosome production both required for viral replication. NS2 is also associated with proteins of ACF and RSF chromatin remodeling complexes. ACF is involved in nucleosome sliding, regulation of DNA replication, and DNA repair [119–121]. For adenovirus, the connection between the adenoviral protein E4orf4 and ACF complex is associated with the regulation of viral gene expression and caspase-independent cell death [122,123]. RSF complex has a role in the reorganization of compact DNA following double-stranded breaks, ensuring

the access for DDR proteins [82]. Altogether, our data demonstrate that CPV NS2 associates with factors of chromatin-modifying complexes. In this regard, NS2 has a clear potential role in chromatin remodeling. However, the mechanisms by which NS2 either directly or indirectly contributes to the reorganisation of chromatin warrant additional studies.

Our BioID data links CPV NS2 to proteins of DDR machinery (Fig 4B) such as MDC1, a transducer of DNA damage signal. MDC1 is an upstream DDR factor, which is recruited to double-strand break sites following the phosphorylation of H2AX by ATM protein kinase [40]. MDC1 has been shown to control the formation of double-strand break foci, which are characterized by the accumulation of γ-H2AX and additional DDR factors [124,125]. In MVM infection, viral replication centers are established adjacent to damage response foci containing both MDC1 and γ-H2AX [14,34]. MVM NS1 has a potential role during the localization of the viral genome to a DNA damage site [109]. For example, NS1 interacts with casein kinase 2 alpha (CK2α), which is required for the recruitment of MDC1 to the chromatin damage site [126,127]. Notably, MVM NS2 has been previously linked to DDR. Studies with NS2-deficient mutant of MVM demonstrated that NS2 is involved in the activation of ATR but not ATM-mediated signaling pathway, and it has a role in H2AX phosphorylation and formation of replication centers [14]. In contrast to MVM, CPV NS2 association with MDC1, a mediator of the interaction between H2AX and ATM [128], suggests that in CPV infection NS2 participates in the DDR signaling pathway which is orchestrated by ATM kinases. Moreover, NS2 is most likely involved in the recruitment of H2AX and MDC1 to the replication region, which is required for the normal progression of DNA damage response [14] essential for the development of the replication centers [14].

Additionally, our BioID data revealed NS2 association with certain downstream factors of DDR from the nucleosome remodeling complex FACT (Fig 4B). The nucleolar histone chaperone FACT is involved in chromatin remodeling and in maintaining chromatin integrity by nucleosome disassembly and reassembly during DNA transcription, replication, and repair [129–133]. In addition, chromatin transcription by RNA polymerases and replication by DNA polymerases are facilitated by FACT [134–136]. Beyond its role in promoting transcription and replication, FACT has also been linked to repression of transcription initiation from some specific cellular promoters, for example during perinuclear positioning and stabilization of heterochromatin [137–139]. During replication stress FACT complex is utilized to substitute γ-H2AX with macro-H2A.1 close to DNA damage regions [140]. Moreover, FACT appears to be involved in repressing transcription initiation of viral promoters during their latent life cycle. In human immunodeficiency virus type I (HIV-1) infection, interaction of FACT proteins with histones H2A/H2B lead to suppressed viral replication and facilitation of viral latency [141]. FACT also binds to an immediate early promotor of human cytomegalovirus and facilitates transcription of human cytomegalovirus immediate early genes [142]. Notably, in lytic HSV-1 infection FACT is localized to the viral replication compartment [30,143]. FACT is recruited by the HSV-1 immediate early protein ICP22 to allow efficient viral transcription at late stages of viral infection [144]. It is evident that CPV NS2 interaction with FACT may have many important functions during the infection, considering that parvoviral genes are transcribed by the FACT-associated cellular RNA polymerase II [134,145].

The complex nuclear chromatin environment presents a challenge for the replication of incoming viral genome. To ensure a successful infection cycle, viruses manipulate a range of cellular chromatin-modifying proteins to promote their own replication and to control cellular anti-viral defense mechanism. Our results demonstrate that NS2 is associated with certain key factors in chromatin remodeling, presumably either to facilitate viral replication and to inhibit cellular functions from reducing virus replication. NS2 interaction with DNA damage sensing and repair machinery is most likely required for the development of viral replication centers,

and could also be involved in the promotion of viral gene transcription, and the transcriptional silencing of host cell chromatin. Our analyses provide unique insights into the potential mechanisms and associations in nuclear processes by which parvoviral NS2 operates in infection.

## Materials and methods

### Cells and viruses

Norden laboratory feline kidney (NLFK) cells and human cervical carcinoma HeLa cells were grown at 37˚C in 5% $CO_2$ in Dulbecco´s Modified Eagle Medium supplemented with 10% fetal calf serum, 1% penicillin/streptomycin, and 1% L-glutamine (Gibco, Thermo Fischer Scientific, Waltham, MA). Flp-In T-REx 293 cell line (R78007, Invitrogen, Life Technologies, Waltham, MA) used for the generation of tetracycline-inducible stable cell lines expressing the HA-BirA*-NS2 were grown at 37˚C in 5% $CO_2$ in Dulbecco´s Modified Eagle Medium supplemented with 10% fetal calf serum, 1% penicillin/streptomycin and 1% L-glutamine. CPV type 2 virus was derived from an infectious plasmid clone p265 by transfection of NLFK cells. Finally, we used two NS2 mutant infectious clones with termination codons in the NS2 coding sequence which led to inactivation of splice donor (G533A) or splice acceptor (A2033T) [6].

### Plasmids

N- and C-terminal fluorescent fusion mammalian expression constructs of CPV NS2 were generated. NS2 gene was PCR amplified from cells transfected with p265 plasmid [146] according to reported donor and acceptor splice sites [6]. For N-terminal fusion, NS2 gene was cloned between EcoRI and BamHI restriction enzyme sites of pEGFP-N3 vector (Clontech), and for C-terminal fusion between BglII and HindIII sites of pEGFP-C1 (Clontech). N-terminal and C-terminal NS2 clones were named NS2-EGFP and EGFP-NS2, respectively. Correctness of all clones was confirmed by sequencing. To determine the possible effect of fluorescent protein position on the cellular distribution of the fusion protein, the cellular localizations of expressed NS2-EGFP and EGFP-NS2 constructs were analyzed in HeLa ells. Both constructs showed similar nuclear distribution including accumulation to the nuclear membrane and nucleolar foci.

NS2 gene surrounded by full AttB sites for gateway cloning was generated from EGFP-NS2 by PCR. The insert was then cloned into pDENTR221 entry vector via gateway BP reaction, and into MAC-tag-C [68] via LR reaction. The resulting NS2-MAC plasmid was used for transfecting Flp-In TREX 293 cells.

### RT-qPCR

The timing of NS1 and NS2 transcription in infected NLFK cells was studied with quantitative Taqman RT-PCR. Cells were plated on 3.5 cm culture dish and infected with CPV ~24 h after plating. To quantify the levels of NS1/NS2 mRNAs, cells were collected in TRIZOL (Gibco) at the selected time points at 4–24 hpi and treated with DNaseI (Promega, Madison, WI). NS1 primers were designed to amplify a 106 bp region of CPV genome that is unique to NS1 mRNA (1126 to 1231; primer sequences: RT-NS1-5, AAATGTACTTTGCGGGACTTGG 1126, RT-NS1-3 CACCTCCTGGTTGTGCCATC 1231). NS2 5'-primer contained sequence from both sides of a splice site (520–530 and 2003–2013; RT-NS2-5, CTCGCCAAAAAGTTG CAAAGAC 520–530 + 2003–2013, RT-NS2-3 TGCAAGGTCCACTACGTCCG, 2075–2094) which are joined in NS2 mRNA. NS2 3' primer was designed to yield a final product of 103 bp. Quantitative RT-PCRs were produced as triplicates. SYBR Green (Applied Biosystems, Foster City, CA) was used for detection. Single dissociation curve (Tm for NS1 = 75˚C and Tm for

NS2 = 79˚C) and AGE-gel confirmed the presence of only one product. Samples were amplified in ABI PRISM 7700 Sequence Detection System (Applied Biosystems) under the following conditions: 2 min at 50˚C and 10 min at 95˚C, followed by 40 cycles of 15 s at 95˚C and 1 min at 60˚C. As an endogenous control for the amount of template cDNA, 18S ribosomal RNA was amplified using TaqMan Ribosomal RNA Control Reagents (Applied Biosystems). Relative gene expression was calculated by comparing the expression level of the target gene to 18s rRNA expression level. Noninfected NLFK cells used as a control did not give any signal. Student's t-test was performed to compare NS1 and NS2 transcription.

## BioID

BioID experiments analyzing NS2 interactome on a protein level were conducted by expressing HA- and BirA*-tagged NS2 construct (MAC tag, [68] named HA-BirA*-NS2 in Flp-In T-REx 293 cells). BioID coupled with mass spectrometry (MS) was performed in biological duplicates on NS2 bait alongside a set of negative controls. Expression of HA-BirA*-NS2 was induced with tetracycline. Cells were treated with 2 μg/ml of tetracycline (Merck KGaA, Darmstadt, Germany) and 50 mM biotin (Merck KGaA) for 20 hours prior to harvesting. Negative controls were treated only with tetracycline. For infected cells both tetracycline and biotin were added at 4 hpi in order to reach 20 h induction. Samples were purified with affinity columns, digested, desalted, and analyzed with Q-Exactive mass spectrometer as described previously [68]. Data was analyzed with SAINTexpress [147], and filtered using a protein-level Bayesian FDR of 0.05 as the cutoff. Controls included a total of 16 runs of four different GFP-BirA* constructs, including NLS-GFP, which should tag nuclear pore components encountered by most cargo [68]. Further filtering was done by integrating control experiment data from CRAPome contaminant repository [148]. Due to the comprehensive panel of control experiments used for filtering, the vast majority of identified cytoplasmic proteins in the NS2 data were filtered out. Prey proteins seen in at least 20% of CRAPome experiments were discarded, unless their average spectral value was over 3 times larger than the repository average for the prey. The resulting high-confidence interactor set was used for further analyses. Protein ANalysis THrough Evolutionary Relationships (PANTHER) classification system (http://www.pantherdb.org/) was used to create GO biological process annotation chart of NS2 interactors (S2 Table).

## MS microscopy

MS-microscopy online tool [68], (www.biocenter.helsinki.fi/bi/protein/msmic/ and proteomics.fi) was used with the filtered data to obtain context-based estimates of NS2 localization. MS microscopy is a tool used to infer the localization of the bait protein based on its interactors. It assumes that the longer a bait protein is around a certain prey, the better that prey is seen in BioID results, and thus by comparing the spectral values of the identified high confidence interactome to those of cellular localization marker proteins, the functional localization of NS2 can be deduced.

## Confocal microscopy

For microscopy, cells were grown on glass cover slips, fixed with 4% PFA and permeabilized with 0.1% Triton X-100 in PBS supplemented with 0.5% BSA. HeLa cells were transfected with either NS2-EGFP or with CPV wt infectious plasmid p265 or with the NS2 splice donor (G533A) or splice acceptor (A2033T) deletion mutants. Transfections were performed either with Lipofectamine 3000 transfection reagent (Thermo Fisher Scientific) or JetOptimus (Polyplus transfection). For immunolabeling studies cells were labeled with antibodies against NS1

(MAb obtained from Caroline Astell, University of British Columbia, Vancouver, CAN) (Yeung et al. 1991), NS2 (mouse Ab) (Wang et al. 1998), HA tag (rabbit polyclonal antibody, rAb; ab9110, Abcam, Cambridge, UK), SMARCA5 (rAb, ab183730, Abcam), macro-H2A.1 (mH2A.1 protein, rAb, ab183041, Abcam), SSRP1 (rAb, ab137034, Abcam), γ-H2AX (either with recombinant Alexa Fluor 647 H2A.X anti-gamma phospho S139 rabbit monoclonal antibody, ab195189 or anti-gamma H2A.X phospho S139 rabbit polyclonal antibody, ab2893, Abcam), H39me3 (rAb, ab8898, Abcam), H3K27ac (rAb, ab4729, Abcam) and MDC1 (rAb, PA5-82270, Thermo Fisher Scientific, Waltham, MA, USA). The primary antibodies were followed by goat anti-rabbit or anti-mouse Alexa 488, 546, 555 and 647-conjugated secondary antibodies (Thermo Fisher Scientific). DNA was labelled either with DAPI or with NucBlue.

The immunolabeled cells were imaged either using Olympus FV laser scanning confocal microscope with UPLSAPO 60x oil immersion objective (NA 1.35), Nikon A1R laser scanning confocal microscope (Nikon Instruments Inc., Tokyo, Japan) with CFI Plan Apo VC 60xH oil immersion objective (NA 1.4) or Leica TCS SP8 FALCON laser scanning confocal microscope (Leica microsystems, Mannheim, Germany) with HC PL APO CS2 63x/1.40 oil immersion objective (NA 1.4). For Olympus the following microscopic parameters were used. DAPI (Invitrogen) was excited with 405 diode laser and 425/50 nm band pass filter was used to detect the fluorescence. Alexa 546 and Alexa 633 were excited with 543 and 633 He-Ne lasers, and the fluorescence were collected with 555–625 nm band pass filter and 647 nm long-pass filter, respectively. Microscopic parameters for Nikon A1R were the following. NucBlue or DAPI (Invitrogen) was excited with a 405 nm diode laser and the fluorescence was detected with a 425–475 nm band-pass filter. EGFP was excited with a 488 nm argon laser and the fluorescence was collected with a 515/30 nm band-pass filter. A 561 nm sapphire laser was used to excite Alexa 546, and the fluorescence was imaged with a 595/50 nm band-pass filter. Stacks of 512 x 512 pixels were collected with a pixel size of 60 nm/pixel in the x- and y- directions, and 120–300 nm in the z-direction. Microscopy images with Leica were acquired as follows: Alexa 546 and Alexa 647 were excited with 557 nm and 653 nm wavelengths, respectively, of pulsed white light laser (80 MHz). The Emission detection range was 568–648 nm for Alexa 546 and 663–776 nm for Alexa 647. DAPI and NucBlue were excited with a 405 nm diode laser and emission between 415–495 nm was detected. Laser powers for all channels were fixed. Image stack size was 512 x 512 with a pixel size of 60 nm/pixel in the x- and y- directions, and a step size of 200 to 300 nm in the z-direction.

Fluorescence intensities in the nucleus were calculated by first segmenting the nuclei by automatically calculating an intensity threshold for the DNA label using Otsu's method [149]. The pixel intensity values within the segmented nuclei were then summed to get the total nuclear intensity. The intensity with respect to the nuclear border was calculated in 2D using the confocal layer with the largest nuclear cross-sectional area. First, the shortest Euclidean distance to the nuclear border for each pixel was calculated. The distance values were then sorted into 2-pixel wide bins, and the mean pixel intensity of each bin was calculated. The mean and standard error values were calculated over the cells.

## Proximity ligation assay

For PLA, HeLa or NLFK cells were either transfected with NS2-EGFP using Lipofectamine 3000 transfection reagent (Thermo Fisher Scientific) or infected with wt CPV. Positive controls were infected with CPV whereas negative controls were noninfected. Technical probe controls were transfected with NS2-EGFP and probed either without antibodies or only with GFP antibody. The cells were fixed 20 h after transfection or 24 h after infection with 4% PFA in PBS and permeabilized with 0.1% Triton X-100 in PBS. The assay was performed with

Duolink In Situ Orange Mouse/Rabbit kit (Merck KGaA). Antibodies used for the PLA analyses were anti-GFP (9F9.F9, ab1218, Abcam) paired with each of the following antibodies SMARCA5 (ab183730, Abcam), macro-H2A.1 (ab183041, Abcam), and SSRP1 (ab137034, Abcam). PLA for infected cells was performed with anti-NS2 antibody paired with each of the following antibodies SMARCA5 (ab183730, Abcam), macro-H2A.1 (ab183041, Abcam), SSRP1 (ab137034, Abcam) and MDC1 (rAb, PA5-82270, Thermo Fisher Scientific). In the control studies the CPV VP2 capsid protein was detected with rabbit polyclonal antibody and intact capsids with a capsid-specific mouse antibody. PLA probes were incubated on cells for 1.5 h at 37˚C in a humidified chamber followed by ligation and amplification according to the manufacturer's protocol. The nuclei were stained with DAPI. PLA signals formed between NS2-EGFP and associated proteins were detected with Nikon A1R confocal microscope with CFI Plan Apo VC 60x water immersion objective (NA 1.2). Cells expressing NS2-EGFP were identified and PLA foci were imaged using excitation with a 561 nm sapphire laser, and the fluorescence was collected with a 595/50 nm band-pass filter. DAPI was excited with a 405 nm diode laser, and the fluorescence was collected with a 450/50 nm band-pass filter. The imaged stacks of single cells were of size 512 by 512 pixels with a pixel size of 60 nm and the z sampling distance of 180 nm. PLA signals in infected NLFK cells were detected with Olympus FV laser scanning confocal microscope with UPLSAPO 60x oil immersion objective (NA 1.35). Infected cells were identified by marginalized chromatin and PLA foci were imaged using 543 He-NE laser for excitation and 555–655 nm band pass filter for fluorescence collection. DAPI was excited with a 405 nm diode laser, and the fluorescence was collected with 425–475 nm band pass filter. The imaged stacks of single cells were of size 512 by 512 pixels with a pixel size of 60 nm and the z sampling distance of 180 nm.

To analyze the number of nuclear PLA signals, the nuclei were segmented from DAPI images by using automatic minimum cross entropy segmentation [150] and by filling any holes that were left inside the segmented regions. The PLA signals were segmented automatically using the maximum entropy algorithm [151]. The geometric centers of the PLA signals were calculated, and the number of signals having their geometric center inside the segmented nucleus was calculated. Because the axial resolution of confocal microscope is significantly lower than the lateral resolution, the PLA signals located near the top and bottom surfaces of the NE cannot be reliably assigned to the nucleus or cytoplasm. For this reason, only signals that had the z-component of their geometric center located within 1.75 um from the z-component of the geometric center of the nucleus were accepted into analysis. It was visually confirmed that this selection excluded signals near the top and bottom surfaces of the NE from the analysis. Poisson regression model [103] was used to evaluate the statistical significance of the number of PLA signals in comparison to the negative control.

## Supporting information

**S1 Fig. Nuclear distribution of NS2-EGFP and NS2 associated proteins.** Representative confocal microscopy images showing the localization of NS2-EGFP (green) together with selected BioID hits SMARCA5, MDC1, macro-H2A.1, and SSRP1 (magenta) in HeLa cells at 24 hpt with NucBlue stained nuclei (blue). Merged images show the localization of NS2-EGFP with the BioID-identified interactors. Close-up images of selected regions are shown (right). Scale bars, 3 μm.
(TIF)

**S2 Fig. Nuclear distribution of NS2 and NS2-associated proteins in infected cells.** Representative confocal microscopy images showing the localization of NS2 (green) identified by NS2 antibody together with selected BioID hits including SMARCA5, MDC1, γ-H2AX,

macro-H2A.1, and SSRP1 (magenta) in NLFK cells at 24 hpi with DAPI-stained nuclei (blue). Merged images show the localization of NS2 with the NS2 interactors. Scale bars, 3 μm.
(TIF)

**S3 Fig. Nuclear distribution of NS1 and NS2 associated proteins in CPV-infected cells.** Representative confocal microscopy images showing the localization of NS1 (red) together with selected NS2 BioID hits including SMARCA5, Macro-H2A.1 and SSRP1 (green) in Nuc-Blue-labeled nucleus (blue) of wt CPV-infected in HeLa cells at 24 hpi. Merged images show the localization of replication protein NS1 with the BioID-identified interactors of NS2. Scale bars, 5 μm.
(TIF)

**S4 Fig. Technical controls and segmentation of the nucleus in PLA assay.** Images of technical controls with either (**A**) both PLA probes without antibodies or (**B**) both PLA probes with only GFP antibody. Representable confocal microscopy sections show PLA signals (green) and the nuclei of HeLa cells stained with DAPI (gray). (**C**) The image shows confocal sections of nuclear DNA stained by DAPI (gray, left) and PLA signals (white, middle). The segmented nucleus is shown by xy- and yz-slices (white, right) in addition to PLA signals (black). The yz-slice is taken along the line shown in red color. Scale bars, 5 μm.
(TIF)

**S5 Fig. Intensity and distribution of DNA in cells transfected with NS2 mutants.** (**A**) Total fluorescence intensities of DAPI-stained DNA in nontransfected, wt and NS2 mutants transfected HeLa cells (24 hpt) (n = 27). (**B**) Localization of DNA as a function of increasing distance from the NE in transfected cells (n = 29). The error bars show the standard error of the mean. Statistical significances were determined using Dunnett's multiple comparison test. The significance values shown are denoted as ** ($p < 0.01$), * ($p < 0.05$) or ns (not significant).
(TIF)

**S6 Fig. Nuclear intensity and distribution of heterochromatin.** (**A**) Representative confocal images of heterochromatin marker H3K9me3 (magenta), NS1 (green), and DAPI (gray) distribution in nontransfected HeLa cells, and cells transfected with wt, NS2 donor and acceptor mutants at 24 hpt. (n = 25). (**B**) Fluorescent intensities and (**C**) nuclear localization of H3K9me3 in nontransfected and transfected cells. The error bars show the standard error of the mean. Statistical significances were determined using Dunnett's multiple comparison test. The significance values shown are denoted as ** ($p < 0.01$), or ns (not significant). Scale bars, 5 μm.
(TIF)

**S1 Table. BioID hits linking NS2 to chromatin organization and mRNA processing.** The table presents 122 NS2-binding proteins that were detected as high-confidence (BFDR $\leq 0.05$) interactors. Two of the inherently distinct groups of NS-associated proteins representing chromatin modification (blue) and DNA damage response (gray) are highlighted. Interactions in the presence or absence of infection are shown.
(XLSX)

**S2 Table. Biological pathways of NS2 BioID hits.** GO functional annotation chart of CPV NS2 high-confidence interactors created by PANTHER classification system for GO Biological process overrepresentation (default FDR < 0.05 filter). The groups are organized by the most specific subterm first, level 0 being the most specific. GO terms for chromatin modification (blue) and DNA damage response (gray) are shown.
(XLSX)

**S3 Table. Unfiltered NS2 BioID data.** Unfiltered SaintExpress-analyzed BioID data used in this study. For each identified prey protein, the average spectral count in infected and noninfected samples are shown, together with BFDR values that are later used for filtering the dataset. Interactors representing four distinct functional groups are colored accordingly: chromatin modification (blue) and DNA damage response (gray).
(XLSX)

**S1 Data. Original data used for analyses.**
(XLSX)

## Acknowledgments

We thank Anssi Mähönen and Lassi Palmujoki for their assistance in constructing BirA-tagged NS2 constructs. Our special thanks go to Wendy Weichert for her help with CPV NS2 mutants.

## Author Contributions

**Conceptualization:** Maija Vihinen-Ranta.

**Data curation:** Salla Mattola, Vesa Aho.

**Formal analysis:** Salla Mattola, Kari Salokas, Vesa Aho.

**Funding acquisition:** Maija Vihinen-Ranta.

**Investigation:** Salla Mattola, Kari Salokas, Elina Mäntylä, Sami Salminen, Satu Hakanen, Minna Kaikkonen-Määttä.

**Methodology:** Salla Mattola, Kari Salokas, Vesa Aho, Elina Mäntylä, Sami Salminen, Einari A. Niskanen, Julija Svirskaite, Teemu O. Ihalainen, Kari J. Airenne, Minna Kaikkonen-Määttä, Colin R. Parrish, Markku Varjosalo.

**Project administration:** Maija Vihinen-Ranta.

**Resources:** Minna Kaikkonen-Määttä, Colin R. Parrish, Markku Varjosalo, Maija Vihinen-Ranta.

**Software:** Vesa Aho.

**Supervision:** Vesa Aho, Kari J. Airenne, Maija Vihinen-Ranta.

**Visualization:** Salla Mattola, Sami Salminen, Einari A. Niskanen, Teemu O. Ihalainen.

**Writing – original draft:** Salla Mattola, Kari Salokas, Vesa Aho, Sami Salminen, Satu Hakanen, Julija Svirskaite, Colin R. Parrish, Maija Vihinen-Ranta.

**Writing – review & editing:** Salla Mattola, Kari Salokas, Vesa Aho, Satu Hakanen, Colin R. Parrish, Maija Vihinen-Ranta.

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
