## [Decision Letter · Decision Letter 0]

7 Oct 2021

Dear Adjunct professor Vihinen-Ranta,

Thank you very much for submitting your manuscript "Parvovirus nonstructural protein 2 interacts with chromatin-regulating cellular 1 proteins" for consideration at PLOS Pathogens. As with all papers reviewed by the journal, your manuscript was reviewed by members of the editorial board and by several independent reviewers. In light of the reviews (below this email), we would like to invite the resubmission of a significantly-revised version that takes into account the reviewers' comments.

The reviewers all agree that this is an interesting and well done study that identifies a number of potentially informative interactions. However, they also all agree that the study still remains descriptive and does not go very far towards demonstrating functional relevance.  The impact would be increased by direct demonstration that the interactions and effects are relevant for virus infection.

We cannot make any decision about publication until we have seen the revised manuscript and your response to the reviewers' comments. Your revised manuscript is also likely to be sent to reviewers for further evaluation.

Sincerely,

Matthew D Weitzman, Ph.D.

Guest Editor

PLOS Pathogens

Karl Münger

Section Editor

PLOS Pathogens

Kasturi Haldar

Editor-in-Chief

PLOS Pathogens

orcid.org/0000-0001-5065-158X

Michael Malim

Editor-in-Chief

PLOS Pathogens

orcid.org/0000-0002-7699-2064

This is an interesting and well done study that identifies a number of potentially informative interactions. But the study does not go very far towards demonstrating their functional relevance and is therefore limited in its impact in the current form.

Reviewer's Responses to Questions

**Part I - Summary**

Reviewer #1: The role of the NS2 proteins of the autonomous parvoviruses have proved to be an enigma. This is especially true for the NS2 protein of the canine parvovirus (CPV). This manuscript makes a number of interesting associations that point to potential CPV NS2 effects on cellular chromatin modifying and DNA damage response proteins. Specific interactions between these cellular proteins and NS2 are identified, and changes in the localization of expressed proteins following infection are observed; however, direct demonstration that these properties of NS2 play a role during infection, which seems to be the aim of the manuscript, and certainly is most significant goal, is lacking.

Reviewer #2: Mattola et al. used proximity-dependent biotin identification (BioID) to screen for proteins that associate with canine parvovirus (CPV) NS2 protein in both infected and transfected cells. The assay revealed a potential role for NS2 in chromatin remodeling and DNA damage response. Moreover, transfection of mutant viral genomes with mutations in the splice donor- or acceptor sites of NS2 resulted in altered levels and distributions of heterochromatin and NS1 protein, when compared to cells transfected with the wild-type genome. Specifically, mutation of the splice donor site of NS2 led to formation of mainly small viral replication compartments (visualized by NS1 staining) compared to the large centers seen with the wild-type genome.

The results provide insights into potential novel roles of NS2 in controlling chromatin remodeling and DNA damage response (DDR) which might be necessary for successful viral replication. The data is interesting but largely descriptive. Some interaction partners such as SMARCA5 and SSRP1 identified by BioID were confirmed also by proximity ligation assay (PLA), but only in cells transfected with NS2-EGFP using a GFP antibody not in infected cells. Most of the other interaction partners identified by BioID were not confirmed with any other assay. Also, experiments to investigate whether the interaction of NS2 with some of the proteins have a positive or negative effect on virus infection/replication, other than the size of NS1 compartments, have not been done.

Reviewer #3: Canine parvovirus (CPV) is a remarkably simply autonomous parvovirus. Consequently, the few gene products encoded by CPV must negotiate with the host cell in complex and multifaceted ways. This study focuses on one of the two non-structural proteins of this virus (NS2) with an approach designed to identify significant interacting partners. The study makes excellent use of complementary approaches including proximity labeling methods to identify transient associations by biochemical (proximity-dependent biotinylation identification) and morphological (proximity ligation assay) means as well as visualization of fluorescently tagged proteins. The spectrum of partners in non-infected cells compared to infected cells suggests a shift in interacting with the nuclear envelope to chromatin. This is novel insight that recapitulates what has been known for some key regulators of the DNA-damage response such as p19-Arf. Another class of proteins associated with NS2 that may provide insight into the function of NS2 are chromatin (nucleosome) remodeling activities.

The results are largely descriptive. However, the comprehensive nature of the analysis and the novel insight of an understudied class of viral effector proteins makes the findings significant to investigators interested in the biology of small DNA viruses. Additionally, although CPV is similar to the small parvovirus of mouse, MVM, significant differences in the function of NS2 from CPV and MVM were identified. This adds to the significance of this report and suggests that a systematic comparative analysis of the NS2 of these two viruses as well as among cell types may prove to be informative. The manuscript is very well-written with an exceptionally clear methods section.

I have no fundamental concerns regarding this manuscript. I have, however, identified several minor issues to correct or address and raise several more major points for consideration.

Reviewer #4: Canine Parvovirus (CPV) Non-structural protein 2 (NS2) is a small ancillary protein whose role in infection is poorly understood and under-researched. This paper reports a sophisticated set of proximity analyses of the association of NS2 with host proteins, particularly those involved in chromatin regulation and the DNA damage response. For this the authors induced a version of NS2 fused at its N terminus to a biotinylation domain, BioID, in both CPV-infected and uninfected cells, then isolated the biotin-labeled proteins and identified them by mass spectrometry. This technique identified an extensive array of NS2-associated proteins. They then used proximity ligation to confirm several of these associations.

**Part II – Major Issues: Key Experiments Required for Acceptance**

Reviewer #1: BioID experiments show direct NS2 interactions with cellular proteins involved in, among other things, chromatin organization and the DNA damage response. Proximity-dependent ligation experiments verify some of the most notable interactions, and localization of expressed NS2 shifts during infection. These experiments are well done and provide clues to potential roles for NS2 during infection. However, the authors claim that “our results provide unique insights into novel roles of NS2 in controlling chromatin remodeling and DNA damage response necessary for successful viral replication.” This remains unsubstantiated.

Major criticisms:

Most importantly, it is imperative to examine NS2 and NS1 expression in the donor and acceptor mutants being studied. Donor mutations often lead to cryptic or spurious splicing, and the products generated from the donor mutant NS region are unknown. This mutant may not generate NS2 at all, or perhaps it generates an NS2 fragment, or perhaps it splices in frame into the NS1 coding region generating a chimeric mutant protein. This knowledge is essential for further interpretation. Also, does the splice acceptor mutation also prevent wt NS2 production as might be expected? If so, and as it behaves much like wild-type, what would that say about the requirement of NS2 during infection? I suspect that performing such experiments may be hampered by the lack of available reagents; however, it remains very difficult to draw conclusions about the role of NS2 with these uncertainties.

The experiments looking at NS2 effects related to DDR proteins is confounded by the uncertainty of the mutants used, but in addition, as the authors point out (line 395), by potential differences in mutant viral replication. If this is the case, any effects attributable to NS2 that can be made are at best secondary in nature. Viral replication of parvoviruses induces a DDR and the authors demonstrate that; however, the authors are trying to make the case that NS2’s role directly affecting the DDR plays a gateway role facilitating infection. This is an attractive possibility; however, notwithstanding NS2’s interaction with DDR proteins here identified, these experiments do not illuminate roles in this process beyond NS2’s anticipated effect on replication – and even this is muddied by lack of characterization of the mutants. Determining the levels of replication of the mutants is an important aspect that may help to begin to resolve this issue.

The experiments describing the correlation of NS1 levels with the DDR protein levels is very confusing. Is NS1 being used as a proxy for replication? The argument that differences in correlation show the requirement of NS2 for progression of the DDR is not convincing, especially in the absence of replication data. Also, discordance of the correlation for the splice acceptor mutant seems inconsistent with its ability to replicate well. These results need additional explanation.

Reviewer #2: 1. Fig. 1: Some confocal images are of low quality; for example, the colocalization of NS2 and nucleolin in Fig. 1C is not very convincing. Also, what statistical analysis was used in Fig. 1A?

2. Functional assays to investigate the effect of selected NS2 interactions on virus infection/replication should be performed.

3. Fig. S1: The bright fluorescent spherical dots may be an artefact from NS2-EGFP overexpression as they cannot be found in the infected cells shown in Fig. S2.

4. Line 234: …”(MKI67, BFDR <0.01).” According to Table 1, BFDR is 0.04 (non-infected) and 0.02 (infected).

5. Line 375: “…findings demonstrate changes in the amount of heterochromatin in response to NS2 donor mutant transfection, suggesting that NS2 has a role in chromatin remodeling.” This is not supported by the data. There is a significant difference between wt and non-infected cells but not between wt and the mutants (Fig. 6B).

Reviewer #3: 1. The relatively novel proximity-dependent biotinylation identification or BioID method is powerful. However, it requires the target protein (NS2 in this case) to accept and epitope tag and fusion with the promiscuous bacterial biotin ligase. The proximal targets identified were associated with a significantly modified, “mutant” form of NS2. What is evidence demonstrates that the modified NS2 protein remains functional and localizes in an identical manner to the unmodified protein? Similarly, although the N- and C-terminal fusions of EGFP to NS2 showed similar localizations after transfection, how does this compare to the distribution of the unmodified NS2 protein?

2. The final statement of the results, "These findings imply that NS2 might have a role in viral replication at cellular sites of DDR," implies that the DNA-damage repair proteins are associated with bona fide sites of cellular DNA damage. Since the spurious and widespread activation of DDR proteins may be a hallmark of many virus infections, this statement should be more carefully supported or appropriately modified.

3. The statistical method used to evaluate difference among spots counted by the PLA assay may be inappropriate. Nonetheless, this is unlikely to affect the conclusion or significance. Specifically, Dunnett’s method is an appropriate means of comparing the means of multiple groups against a single control. The comparison to a control group is appropriate. However, a more appropriate analysis for count data would be a Poisson regression or Dunnett’s method applied to suitably transformed count values.

Reviewer #4: While the studies are meticulously performed and extensively controlled, several aspects of this study are puzzling. First is the surprisingly large number of apparent partners of such a small viral protein, only 165 amino acids in length. This may be because the fusion actually triples the molecular weight of NS2, and probably allows the biotinylation domain to interact promiscuously with many proteins in fairly loose proximity to any authentic, directly-interacting partner of NS2. The proximity ligation technique also does not measure direct molecular interactions.

Unfortunately, the authors do not pursue any more biochemical approaches, such as co-immunoprecipitation, to confirm direct, specific binding between NS2 and any of the identified proteins in the study.

Likewise, while the authors present an exhaustive discussion of how these possible interactions might be involved in CPV infection, the biological relevance of any of the identified associations is unclear. Indeed, it is puzzling why such an apparently interactive protein does not seem to play an essential role in CPV infection. For instance, the original paper describing CPV NS2 (ref 6), describes the elimination of the full-length protein by the introduction of in-frame termination codons in the NS2 ORF. This study found no significant effect of knocking out NS2 on the course of CPV infection of either cells in culture or dog tissues in vivo.

**Part III – Minor Issues: Editorial and Data Presentation Modifications**

Reviewer #1: Minor criticisms:

Line 354: The magnitude of these differences are so marginal that it is not reasonable to make this conclusion without corroborating evidence. Also if the acceptor mutant does not make NS2 (and we need to know this), how can such an effect be attributable to NS2?

Line 363 and 365: “acceptor mutant” rather than “acceptor donor mutant”?

It is not always clear which experiments use NLFK cells and which use HeLa. Are there any concerns using HeLa cells to examine NS2/DDR protein connections?

Reviewer #2: 1. Fig. 1C legend should be: …NS2 (green) and nucleolin (red).

2. What cells were used for the BioID assay? Please indicate.

3. Line 318: In what sense did the intensities (spectral counts) correlate with the numbers of PLA foci? How was this determined. Please clarify.

4. Line 604: “…annotation chart of NS2 interactors (Table S1).” Should this be Table S2?

5. In the text and figures non-infected sometimes means NS2-EGFP transfected (e.g. Figs. 2, 3), sometimes it means non-transfected (e.g. Figs. 6, S4, S5, 7). This is confusing.

6. Fig. S2: Nuclear staining is missing.

7. Fig. 5A: the representative image of the macro H2A.1 staining has roughly 70 foci, not 135+/-12 as indicated in the results (line 314).

Reviewer #3: What exactly does the phrase “emergence and distribution of DNA damage effector proteins” mean? Perhaps a more direct statement of the concept could be used in the Abstract instead of this somewhat obscure description.

The phrase in the abstract, “studies with NS2 mutants indicated that NS2 performs functions that affect the quantity and distribution of DNA damage effector proteins” is also vague and conveys little information. Rather than this vague description, perhaps specific and succinct details could be provided.

Another example of unnecessarily vague writing is in the Author Summary. The statement ”serves a previously undefined important function in essential nuclear processes that occur during viral infection” does not convey meaningful specific information. Rather than simply state these vague assertions, tell the reader what they are.

The network graph presented in Fig 2A is nearly indecipherable and of marginal utility. It seems a textual presentation with this information (of less complexity than Table S1) could be more valuable.

Line 175: The studies in MVM (reference 70) are cited as being able to "account for the relatively small amount of nuclear NS2 detected in CPV infection." If this is referring to a CPV infection, more information on the "small amount of nuclear NS2" is required to justify this statement.

Line 187: What is the basis for stating that "many" DDR factors reside in nucleoli? Is it more than the nucleoplasm proper? Qualification of this statement and a reference is needed.

Results of the associations measured by "BioID" are introduce at line 192 with no discussion on the method, its strength and its limitation. It would be appropriate to include some preamble on this uncommon method such as was done for the PLA method on at lines 304-308.

I am curious as to why the strongest interaction detected was with Ki-67? Is there any evidence pointing to the mitotic role of Ki-67 versus the role it plays in interphase cells?

Although informative, the text describing the identification of various DDR factors associated with NS2 (lines 258 - 289) seems to be more of a discussion of the DNA-damage response than a statement of the results. Nonetheless, a simple recitation of associated factors would be of little value. On balance, this is appropriate but could possibly benefit from some editing to reduce the amount of text.

Were any other factors influencing the choice of SMARCA5, macro-H2A.1 and SSRP1 as targets for the PLA other than availability of antibodies. If so, it may be helpful to provide the rationale.

The statement at line 323, "It is possible that NS2 association with SSRP1 is transient as it is detected clearly by BioID whereas hardly by PLA," is likely but it may help the reader to justify this with a reference supporting the notion that transient interactions are better detected by BioID.

Why is the NS2 acceptor mutant virtually wild-type? Are alternative splice acceptors being used? If the use of the NS2 splice acceptor mutant as wild-type is standard practice, it would be helpful to cite a reference for this. Otherwise, an explanation of why this "mutant" is not mutant is appropriate.

The measurement of "correlations" of various proteins localized in Fig. 7 are discussed at line 404 without any definition of what is meant by "correlation." This should be added.

Required corrections a the indicated line number

90: space needed in "thenuclear"

273: "chaperon" is misspelled

343: space needed in "thesplice"

347: the phrase "in the splice donor mutant virus transfected cells" requires hyphens to connect adjectival nouns.

353: "donor mutant was transfected; in contrast it was lower compare" requires editing (a comma, for one thing)

417: Rephrase "This could propose that…"

Reviewer #4: (No Response)

PLOS authors have the option to publish the peer review history of their article (what does this mean?). If published, this will include your full peer review and any attached files.

Reviewer #1: No

Reviewer #2: No

Reviewer #3: No

Reviewer #4: No
---

## [Decision Letter · Decision Letter 1]

24 Feb 2022

Dear Adjunct professor Vihinen-Ranta,

Thank you very much for submitting your manuscript "Parvovirus nonstructural protein 2 interacts with chromatin-regulating cellular proteins" for consideration at PLOS Pathogens. As with all papers reviewed by the journal, your manuscript was reviewed by members of the editorial board and by several independent reviewers. The reviewers appreciated the attention to an important topic. Based on the reviews, we are likely to accept this manuscript for publication, providing that you modify the manuscript according to the review recommendations.

Please address the comments of reviewer 1. Experiments proposed by reviewer 4 do not need to be performed but their comments should be addressed in the text.

Sincerely,

Karl Münger, Ph.D.

Section Editor

PLOS Pathogens

Karl Münger

Section Editor

PLOS Pathogens

Kasturi Haldar

Editor-in-Chief

PLOS Pathogens

orcid.org/0000-0001-5065-158X

Michael Malim

Editor-in-Chief

PLOS Pathogens

orcid.org/0000-0002-7699-2064

Please address the comments of reviewer 1. Experiments proposed by reviewer 4 do not need to be performed but their comments should be addressed in the text.

Reviewer Comments (if any, and for reference):

Reviewer's Responses to Questions

**Part I - Summary**

Reviewer #1: CPV NS2 is an enigma. In this revised manuscript the authors convincingly demonstrate that NS2 interacts with a number of proteins involved in chromatin remodeling and the cellular DNA damage response. This is valuable information but essentially descriptive. The authors analyze mutants of NS2 to make a functional connection; however, questions remain concerning the use of the NS2 mutants chosen.

Reviewer #2: The authors have responded to all reviewer comments and substantially revised/improved the manuscript.

Reviewer #3: This study is a revised submission focused on one of the two non-structural proteins of canine parvovirus (NS2). This work seeks to identify significant interacting cellular partners of NS2. Biochemical proximity labeling methods and semi-quantitative morphological assays support the notion that NS2 shifts from interacting with the nuclear envelope to chromatin with consequences on cellular chromatin compaction.

The revised manuscript appears to have addressed nearly all of the concerns expressed by previous reviewers. Notably the new network graph appears to convey more accessible information and the large amount of text reviewing the DNA damage-response has been appropriately tightened. The tendency to overstate inferences regarding the NS2 protein has be corrected. Appropriate statistical methods have been applied and better confocal images are provided with insightful quantitative analyses. The results remain largely descriptive but provide new information on this understudied class of viral effector proteins. The clarity of the manuscript, which was strong before, has been improved.

As before, I have no fundamental concerns regarding this manuscript. However a limitation of this work that remains is on the nature of proteins being expressed by the NS2 splice donor and splice acceptor mutants. This was a key concern of one previous reviewer. The authors provide a strong argument for the likely absence of NS2 protein expressed by the key splice donor mutant but the possibility of an unexpected (and unidentified) fusion protein remains, which could confound the interpretation of the functional aspects discussed in this paper. Nonetheless, the major contribution of this finding is provide strong evidence for key binding partners of NS2. These binding partners strongly implicate elements of the DNA damage-response and chromatin organization as been likely targets of NS2.

Reviewer #4: This is an extensive, sophisticated and meticulously performed study of interactions between host cell proteins and what the authors represent as the NS2 gene product of Canine Parvovirus (CPV). The major concern that remains for this reviewer is whether these interactions are truly NS2 specific. The reason for this concern comes from a consideration of a previously published analysis of CPV NS2 mutants.

The NS2 gene of CPV encodes a 165 amino acid polypeptide, of which the N-terminal 87 amino acids are contributed by genomic sequence common to NS2 and the N-terminus of NS1, the major replicative non-structural protein of the virus. In NS2, this common N-terminal sequence is linked by mRNA splicing to a further 78 amino acids encoded in an alternative reading frame colinear with downstream sequences of the NS1 gene. The original paper characterizing CPV NS2 (Wang et al., 1998 – Ref 6), describes the analysis of five mutants designed to disrupt the expression of NS2 without affecting the amino acid sequence of NS1. Of these, three translation termination mutants and a splice acceptor mutant showed no discernable phenotype with respect to viral DNA replication, structural protein synthesis, capsid assembly and production of infectious virus, and where tested, infection of dogs, leading to the conclusion that most, if not all, of the C-terminal NS2-specific sequence is not required for any currently measurable facet of the viral infectious cycle.

This is in marked contrast to the results of similar experiments with NS2 mutations in the genomes of rodent parvoviruses, predominantly Minute Virus of Mice, where abrogation of the second exon of NS2 leads to abortive infection in normal cells of the host species and abolishes pathogenesis in infected host animals. One of the five CPV mutations, that affecting the donor splice site, does exhibit a phenotype of impaired viral DNA replication and capsid protein synthesis. This puzzled Wang et al., who stated that “Analysis of the effects of altering the splice donor or acceptor sequences was complicated by the fact that in each case an alternative sequence used to splice the message was detected by RT-PCR.” Despite this warning, and the similar concerns of reviewer 1, the authors continue to emphasize that they consider the splice donor mutant to be a simple NS2 null mutant, with an interesting phenotype representing solely the absence of NS2. It seems extremely unwise to ignore the strong indications from the other mutants studied in the Wang et al., paper that the C-terminal 78 amino acids of NS2 are dispensable, and thus are extremely unlikely to harbor interaction domains of any relevance to the viral life cycle.

This consideration suggests the distinct possibility that the biologically significant interactions revealed in the current study are properties solely of the 87 N-terminal amino acids of NS2 – and therefore also of NS1. Thus, it seems quite possible that the interactions revealed in this manuscript are properties of NS1, or that NS2 acts as an N-terminal domain of NS1 liberated from its nickase, helicase and transcriptional activator functions by mRNA splicing.

**Part II – Major Issues: Key Experiments Required for Acceptance**

Reviewer #1: To determine the biological relevance of the interactions, and the functional significance of NS2 in this regard, the authors rely on analysis of transfected CPV plasmid clones that bear mutations of NS2 which have been previously characterized (Wang, 1998). In those published studies, mutations affecting the R2 splice donor or acceptor, as well as termination mutations (one, at nt 2033, not far downstream of the R2 splice acceptor), were analyzed. None of these generated NS2 molecules recognizable by antibodies directed to the NS2-specific C-terminal exon; however, all showed wild-type phenotypes, except for the splice donor mutant, which was substantially defective. The same splice donor and acceptor mutants were used in this study. As evidence of a specific NS2 connection the authors show that the splice donor mutant displayed differences in the assays examined, while the splice acceptor mutant, also devoid of the NS2-specific exon, did not. It is important to note that the N-terminal exon of NS2 is shared with NS1. How to square those results with the results presented here?

In their rebuttal to previous critique of this point the authors suggest that: “some key functions essential for viral replication are most likely removed from the NS2 splice donor mutant”, and “the NS2 splice donor mutation likely leads to drastically shorter protein than the splice acceptor.” Little is known about what products are actually generated by the splice donor or splice acceptor mutant, and cryptic splicing was detected for both at levels not reported (Wang, 1998). The previously identified termination mutant, however, would be predicted to generate an NS2 molecule truncated merely 10 amino acids following splicing into the NS2-specific C-terminal exon.

There seem to be a limited number of possibilities that would support the authors arguments. One is that the splice donor mutation leads to accumulation of very little R2 at all, and that, although unlikely, a certain amount of the independent NS1/2 shared exon is required for wild-type “NS2” activity. Alternatively, since the 2033 termination mutant was previously shown to be wild-type, perhaps the 10 amino acids left in the NS2 unique exon of that mutant affords its function and is the bit the authors are referring to. (Cryptic splicing in the splice acceptor mutant joins significantly downstream at nt 2099 and has a larger NS2-spcific exon piece.) Finally, it also remains possible that aberrant splicing of the donor mutation (which uses a cryptic donor 4 nts downstream of the wildtype site) results in a dominant negative NS2 molecule responsible for the effects seen.

I suppose the authors' rationale for using these mutants can be accepted – even though the precise nature of their NS2 defects are not clear. But the possibilities mentioned above pose a nagging, obvious, open question coloring the manuscript which need to be more fully discussed somewhere in the manuscript, and hopefully addressed experimentally at a later date.

Reviewer #2: no additional comments

Reviewer #3: No major experiments required

Reviewer #4: The probability that the splice donor and acceptor mutants activate cryptic acceptor and donor splice sites, respectively, should be examined by sequencing of RT-PCR products of appropriately spaced primers. This would reveal whether either of these mutations result in the expression of aberrant forms of NS1.

Alternatively, the possibility that some or all of the observed interactions are properties of the NS1/NS2 common N-terminal domain could be tested by BioID analysis of an N-terminal fusion of the BirA* tag with NS1/2, coupled with a translation termination codon replacing the splice donor site. Showing that this was the case would justify the proposal that these interactions significantly affect the viral life cycle.

**Part III – Minor Issues: Editorial and Data Presentation Modifications**

Reviewer #1: (No Response)

Reviewer #2: no additional comments

Reviewer #3: None noted in primary manuscript.

Reviewer #4: (No Response)

PLOS authors have the option to publish the peer review history of their article (what does this mean?). If published, this will include your full peer review and any attached files.

Reviewer #1: No

Reviewer #2: No

Reviewer #3: No

Reviewer #4: No

Figure Files:

Data Requirements:

Reproducibility:

References:

---

## [Editor Report · Decision Letter 2]

15 Mar 2022

Dear Adjunct professor Vihinen-Ranta,

We are pleased to inform you that your manuscript 'Parvovirus nonstructural protein 2 interacts with chromatin-regulating cellular proteins' has been provisionally accepted for publication in PLOS Pathogens.

Best regards,

Matthew D Weitzman, Ph.D.

Guest Editor

PLOS Pathogens

Karl Münger

Section Editor

PLOS Pathogens

Kasturi Haldar

Editor-in-Chief

PLOS Pathogens

orcid.org/0000-0001-5065-158X

Michael Malim

Editor-in-Chief

PLOS Pathogens

orcid.org/0000-0002-7699-2064
---

## [Editor Report · Acceptance letter]

4 Apr 2022

Dear Adjunct professor Vihinen-Ranta,

We are delighted to inform you that your manuscript, "Parvovirus nonstructural protein 2 interacts with chromatin-regulating cellular proteins," has been formally accepted for publication in PLOS Pathogens.

Best regards,

Kasturi Haldar

Editor-in-Chief

PLOS Pathogens

orcid.org/0000-0001-5065-158X

Michael Malim

Editor-in-Chief

PLOS Pathogens

orcid.org/0000-0002-7699-2064